# Habitizing Diffusion Planning for Efficient and Effective Decision Making

**Haofei Lu** [1]  **Yifei Shen** [2]  **Dongsheng Li** [2]  **Junliang Xing** [1]  **Dongqi Han** [2]

## Abstract

Diffusion models have shown great promise in decision-making, also known as diffusion planning. However, the slow inference speeds limit their potential for broader real-world applications. Here, we introduce **Habi**, a general framework that transforms powerful but slow diffusion planning models into fast decision-making models, which mimics the cognitive process in the brain that costly goal-directed behavior gradually transitions to efficient habitual behavior with repetitive practice. Even using a laptop CPU, the habitized model can achieve an average **800+ Hz** decision-making frequency (faster than previous diffusion planners by orders of magnitude) on standard offline reinforcement learning benchmarks D4RL, while maintaining comparable or even higher performance compared to its corresponding diffusion planner. Our work proposes a fresh perspective of leveraging powerful diffusion models for real-world decision-making tasks. We also provide robust evaluations and analysis, offering insights from both biological and engineering perspectives for efficient and effective decision-making.

## 1. Introduction

The trade-off between computational cost and effectiveness is a key problem in decision making (Kahneman, 2011; Sidarus et al., 2019; Clark et al., 2021). In machine learning, probabilistic generative models have increasingly been used for planning of the outcome of actions (Ha & Schmidhuber, 2018; Hafner et al., 2019; 2023). Recently, diffusion models have been used for decision making (Janner et al., 2021; Ajay et al., 2022; Du et al., 2024), in particular of-

The work was conducted during the internship of Haofei Lu (luhf23@mails.tsinghua.edu.cn) at Microsoft Research Asia [1]Department of Computer Science and Technology, Tsinghua University [2]Microsoft Research Asia. Correspondence to: Junliang Xing <jlxing@tsinghua.edu.cn>, Dongqi Han <dongqihan@microsoft.com>.

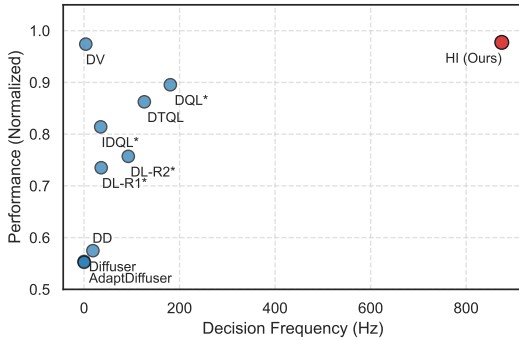

*Figure 1.* **Performance vs. Frequency.** Performance is normalized across MuJoCo, AntMaze, and Kitchen tasks from D4RL. Decision frequency (Hz) is measured on a laptop **CPU** (Apple M2, MacBook). Habitual Inference (HI), a lightweight model generated by our Habi, achieves an optimal balance between performance and speed. See Table 1 for more results.

fline reinforcement learning (RL) (Levine et al., 2020; Fu et al., 2020) by exploiting the generative power of diffusion models for making plans (trajectory of states). While recent diffusion model-based decision makers achieve state-of-the-art performance on standard offline RL benchmarks (Wang et al., 2023; Lu et al., 2025), the computational cost of diffusion models remains a significant challenge – models like diffuser and its variants usually take more than 0.1 seconds, sometimes even more than 1 second, to make a simple decision using a decent GPU (Janner et al., 2021; Liang et al., 2023; Lu et al., 2025), which is unacceptable for real-world applications.

Meanwhile, it has been widely known and researched that humans and animals can make optimal decisions using limited energy. In cognitive science and psychology, decision making is often considered driven together by two kinds of behaviors (Dolan & Dayan, 2013; Wood & Rünger, 2016): a slow, deliberate **goal-directed** one, and a fast, automatic **habitual** one. Goal-directed behavior, focuses on careful planning and precise evaluation of the future outcomes (the process also known as System 2 thinking (Kahneman, 2011)), making it more reliable but time-consuming. In contrast, the habitual behavior (System 1 thinking (Kahneman, 2011)) selects actions in a model-free manner – without considering subsequent outcomes, thus computationally-efficient while could be less reliable.

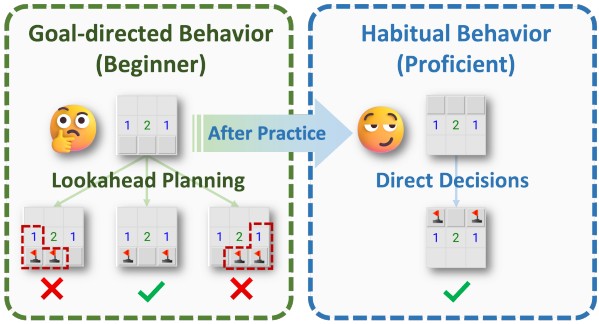

*Figure 2.* **An illustrative example of the process of habitization in playing the Minesweeper game.** With practice, one's decision-making relies less on deliberate goal-directed planning and more on context-dependent habitual behavior.

In this study, we are inspired by one of the findings on habits and goals, known as *habit formation*, or **habitization** (Wood & Rünger, 2016; Han et al., 2024). That is, *the brain will gradually transform slow, deliberate goal-directed behavior into fast, habitual behavior when repetitively doing a task* (see Figure 2 for an illustrative example).

The key reason behind habitization is the hard need for a trade-off between efficiency (decision time and energy) and effectiveness (behavior performance) for animals to survive. A recent study (Han et al., 2024) proposes a computational framework to model the interaction between the two behaviors: As many real-world tasks are few-shot or zero-shot, animals need to utilize their world model to perform goal-directed planning to make reliable decisions (Lee et al., 2014), which is computationally costly. To improve efficiency, they must meanwhile "extract" the goal-directed decision strategy to a habitual decision model that straightforwardly makes decisions without planning by world models.

We notice that diffusion planning (Janner et al., 2022; Ajay et al., 2022) is akin to goal-directed behavior, as both generate plans in the future before making a decision, and both are powerful but slow, hampering their real-world usage. Then, an idea naturally arise: Can we develop a habitization process like in the brain to transform slow, effective diffusion planning into fast, straight-forward habitual behavior?

In this work, we provide a positive answer to this question. We develop a general framework, referred to as **Habi**, that mimics the brain's habitization process. Inspired by Han et al. (2024), Habi leverages variational Bayesian principles (Kingma & Welling, 2014) to connect the slow, yet powerful diffusion planner's policy (as posterior) with a fast, habitual policy model (as prior). By maximizing the evidence lower bound (Kingma & Welling, 2014), the habitual policy model is trained to take advantage of pretrained diffusion planners while maintaining high efficiency, mimicking the habitization process in the brain (Wood & Rünger, 2016).

Habi is featured with the following advantages:

**Efficiency**: The habitized policy model is lightweighted and super-fast, providing orders-of-magnitude speedup over existing diffusion planners (Figure 1).

**Effectiveness**: The habitized policy can compete with the state-of-the-art model that are much slower (Figure 1).

**Versatility**: Habi can be used straightforwardly for any diffusion planning and diffusion policy models.

We further conduct comprehensive evaluations across various tasks, offering empirical insights into efficient and effective decision making.

## 2. Background

### 2.1. Offline Reinforcement Learning.

Offline reinforcement learning (RL) (Fujimoto et al., 2019; Levine et al., 2020) considers the case that an agent learns from a fixed dataset of previously collected trajectories without interacting with the environment. The objective is to train a policy that maximizes the expected return, defined as $\mathbb{E}\left[\sum_{h=0}^{\text{end}} \gamma^h r_{t+h}\right]$, when deployed in the environment. Here, $r_t$ is the immediate reward and $\gamma$ is the discount factor (Sutton & Barto, 1998). Offline RL is particularly relevant in scenarios where exploration is costly, risky, or impractical, requiring agents to maximize the utility of existing data for careful planning. Key challenges include handling high-dimensional (Levine et al., 2020), long-horizon dependencies and deriving near-optimal policies from potentially sub-optimal datasets (Fujimoto et al., 2019). These factors position offline RL as an ideal benchmark for evaluating advanced decision-making methods. In this work, we evaluate our framework using a standard offline RL benchmark D4RL (Fu et al., 2020), providing a rigorous and consistent comparison of its decision-making capabilities against previous baselines.

### 2.2. Diffusion Models for Decision Making.

Diffusion models have recently demonstrated remarkable potential in decision-making tasks due to their ability to model complex distributions. Compared to classical diagonal Gaussian policies (Haarnoja et al., 2018; Schulman et al., 2017; Kostrikov et al., 2021; Kumar et al., 2020), diffusion models have achieved state-of-the-art performance in both online (Wang et al., 2024a; Yang et al., 2023; Psenka et al., 2023; Ren et al., 2024) and offline reinforcement learning (Chen et al., 2023a; Li et al., 2023; Wang et al., 2023; Hansen-Estruch et al., 2023; Janner et al., 2022), as well as demonstration learning (Ze et al., 2024; Chi et al., 2023). There are two main ways diffusion models are applied in decision-making:

(1) *Diffusion planner* (Ajay et al., 2022; Janner et al., 2022; Liang et al., 2023) models a trajectory $\tau$ consisting of the

current and subsequent $H$ steps of states (or state-action pairs) in a horizon:

$$\tau = \begin{bmatrix} s_t, s_{t+1}, \cdots, s_{t+H-1} \end{bmatrix} \text{ or } \begin{bmatrix} s_t, s_{t+1}, \cdots, s_{t+H-1} \\ a_t, a_{t+1}, \cdots, a_{t+H-1} \end{bmatrix}.$$

(2) *Diffusion policy* (Hansen-Estruch et al., 2023; Wang et al., 2023) leverages diffusion models for direct action distribution $p(a_t|s_t)$ modeling, and can be viewed as modeling trajectory $\tau = [s_t \ a_t]$ with planning horizon $H = 1$.

Recent research has expanded the applications of diffusion planning to broader domains, such as vision-based decision-making (Chi et al., 2023) and integration with 3D visual representations (Ze et al., 2024). A recent study (Lu et al., 2025) provides comprehensive analysis of key design choices, offering practical tips and suggestions for effective diffusion planning. Our work, Habi, is orthogonal to diffusion planners or diffusion policies, and can be viewed as an adaptive, general framework to habitize diffusion planning into efficient, habitual behaviors.

### 2.3. Auto-Encoding Variational Bayes

Variational Bayesian (VB) approaches in deep learning have been popular since the introduction of the variational autoencoder (VAE) (Kingma & Welling, 2014; Sohn et al., 2015). The core idea is to maximize the evidence lower bound (ELBO) of an objective function of a probabilistic variable $x$ so that we can replace the original distribution with a variational approximation based on a latent variable $z$ (Alemi et al., 2017). The ELBO can be written as:

$$\text{ELBO} = \mathbb{E}_{z \sim q(z)} \log P(x(z)) - D_{\text{KL}} \left[ q(z) || p(z) \right], \quad (1)$$

where $p, q$ indicates the prior and posterior distributions of $z$, respectively. $\log P(x(z))$ is the log-likelihood of correctly reconstructing data $x$ from $z$, and $D_{\text{KL}}$ denotes Kullback–Leibler (KL) divergence (Kullback & Leibler, 1951).

An important property of ELBO is that although the log-likelihood term in Equation 1 is calculated with posterior samples $q(z)$, the likelihood over prior distribution $p(z)$ is also optimized with the KL-divergence term (see Appendix B). Therefore, it is possible to reconstruct the data $x$ using prior $p(z)$ and the decoder $x(z)$ after training.

## 3. Methods

### 3.1. The Bayesian Behavior Framework

The Bayesian behavior framework (Han et al., 2024) provides a mathematical model for the interaction between habitual and goal-directed behaviors. A latent Bayesian variable $z$ encodes the two behaviors with its prior and posterior distributions, respectively:

$$\text{habitual action} \leftarrow z^{\text{prior}},$$
$$\text{goal-directed action} \leftarrow z^{\text{post}}.$$

The intuition behind such formation is that posterior distribution in Bayes theory relies on additional information than the prior. Habitual behavior relies simply on context (current state), thus encoded as prior; whereas goal-directed behavior is refined by additional evidence (current state + plan of future states), thus encoded as posterior. The interplay between two behaviors (including the habitization process) can be modeled by minimizing a free energy function (Friston et al., 2006) (mathematically equal to the negative of ELBO (Kingma & Welling, 2014), also known as the deep variational information bottleneck (Alemi et al., 2017)):

$$\mathcal{L} = \underbrace{\mathbb{E}_{q(z)} \left[ \text{Recon. loss} + \text{RL loss} \right]}_{Accuracy} + \underbrace{D_{\text{KL}} \left[ q(z) || p(z) \right]}_{Complexity},$$

where $q(z)$ and $p(z)$ denotes the posterior and prior probabilistic density function of $z$, respectively. Recon. and RL loss (accuracy) corresponds to learning the state-transition model and policy improvement, notably over the expectation using posterior $z$. In the habitization process, the KL-divergence (complexity) term can be intuitively understood as aligning habitual behavior with the goal-directed one (Appendix B).

The Bayesian behavior framework was shown to replicate key experimental findings in cognitive neuroscience (Dolan & Dayan, 2013; Wood & Rünger, 2016), in an online RL setting. However, Han et al. (2024) used simple T-maze navigation tasks (O'Keefe & Dostrovsky, 1971; Olton, 1979) to compare with neuroscience experiments, and their methods have not been compared with state-of-the-art models from the machine learning community.

### 3.2. Habi: A Framework to Habitize Diffusion Planners

Inspired by Han et al. (2024), we aim to solve the efficiency problems of diffusion planners by developing a framework, which we call **Habi**, to convert slow, careful diffusion planning into fast, habitual actions, while keeping the effectiveness and the probabilistic nature of policy. Habi consists of two accordingly stages: **(a)** Habitization (Training) and **(b)** Habitual Inference, as depicted in Figure 3.

Habi aligns the decision spaces of habitual behaviors and deliberate goal-directed (diffusion model-based) planning. The habitual and goal-directed behaviors are encoded as the prior and posterior distributions of a latent probabilistic variable $z$, respectively.

As powerful diffusion planner models have already been developed by existing studies (Lu et al., 2025), we can leverage the off-the-shelf, pretrained diffusion planning models.

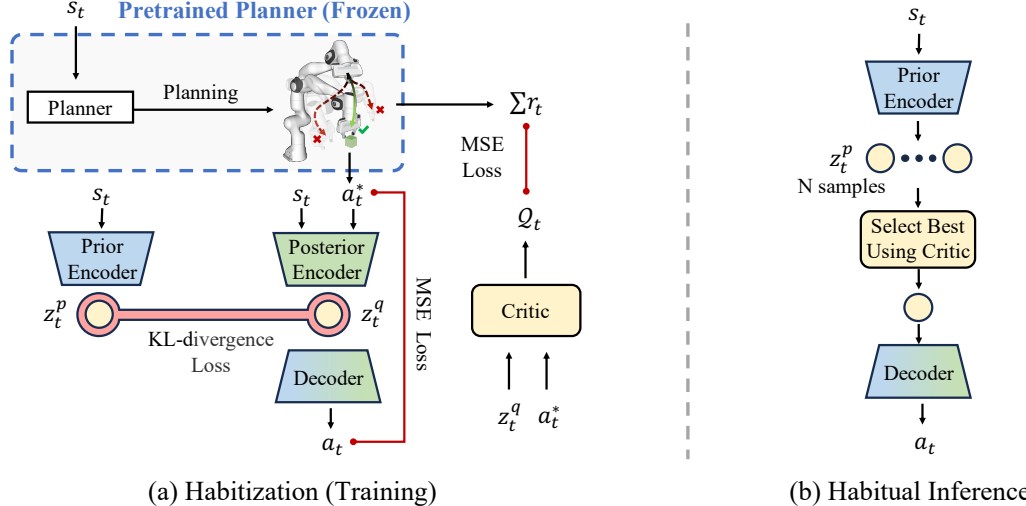

(a) Habitization (Training)  (b) Habitual Inference

*Figure 3.* **The diagram of Habi. (a) During the Habitization (Training) stage**, Habi learns to reconstruct actions from plans generated by a diffusion planner, with the decision spaces of habits (prior) and planning (posterior) aligned via KL divergence in the latent space. Trainable parts include Prior Encoder, Posterior Encoder, Decoder, and Critic. **(b) During the Habitual Inference (HI) stage**, only the lightweight prior encoder and latent decoder are required, enabling fast, high-quality habitual behaviors for decision-making.

### 3.3. Transition from Goal-directed to Habitual Behavior

In this section, we will detail how a behavior policy is *habitized* from a given diffusion planner (Figure 3a). In our Bayesian behavior framework, the Bayesian latent variable $z_t^p$ (or $z_t^q$, where $p$ denotes prior, and $q$ indicates posterior) is treated as a random variable following diagonal Gaussian distribution $\mathcal{N}(\mu_t^p, \sigma_t^p)$ (or $\mathcal{N}(\mu_t^q, \sigma_t^q)$). The subscript $t$ denotes the timestep as we consider a Markov decision process (Bellman, 1957). For simplicity, we will omit the subscript $t$ for the following formulations.

The prior distribution $(\mu_p, \sigma_p)$ is obtained from a mapping (feedforward network) from state $s$ (Figure 3a):

$$z^p \sim \mathcal{N}(\mu^p, \sigma^p) \leftarrow \text{PriorEncoder}(s). \quad (2)$$

The posterior $z^q$ is trained to encode the goal-directed behavior by auto-encoding the diffusion planner's action $a^*$ (Figure 3a).

$$z^q \sim \mathcal{N}(\mu^q, \sigma^q) \leftarrow \text{PosteriorEncoder}(s, a^*). \quad (3)$$

A reconstruction loss $\mathcal{L}_{\text{recon}}$ is introduced (We provide implementation details in Appendix Appendix C.3):

$$\mathcal{L}_{\text{recon}} = \left\lVert \text{Decoder}(z^q) - a^* \right\rVert_2. \quad (4)$$

Intuitively, to make habitual behaviors consistent with goal-directed decisions, the decision spaces of the prior and posterior distributions are aligned under a constraint by Kullback-Leibler (KL) divergence (Kullback & Leibler, 1951):

$$\mathcal{L}_{\text{KL}} = D_{\text{KL}}\big[q(z|s, a^*)||p(z|s)\big] \quad (5)$$

$$= \log \frac{\sigma^p}{\sigma^q} + \frac{(\sigma^q)^2 + (\mu^q - \mu^p)^2}{2(\sigma^p)^2} - \frac{1}{2}, \quad (6)$$

where $q(z|s, a^*)$ is the posterior distribution representing goal-directed behaviors, and $p(z|s)$ is the prior distribution representing habitual behaviors.

The overall habitization loss is elegantly defined as (which is the famous (negative of) ELBO with adjusted KL weighting (Higgins et al., 2017)):

$$\mathcal{L} = \mathcal{L}_{\text{recon}} + \beta_{\text{KL}}\mathcal{L}_{\text{KL}}, \quad (7)$$

where $\beta_{\text{KL}}$ is a weighting factor balancing reconstruction accuracy and decision space alignment. $\beta_{\text{KL}}$ may vary greatly across tasks, and a small $\beta_{\text{KL}}$ may lead to poor alignment, while a large $\beta_{\text{KL}}$ may result in poor reconstruction (Higgins et al., 2017; Alemi et al., 2017). To enable habitization with minimal human intervention, Habi used an adaptive $\beta_{\text{KL}}$ mechanism (Haarnoja et al., 2018; Han et al., 2022) that dynamically adjusts its value during training:

$$\mathcal{L}_{\beta_{\text{KL}}} = \log \beta_{\text{KL}} \cdot (\log_{10} \mathcal{L}_{\text{KL}} - \log_{10} D_{\text{KL}}^{tar}), \quad (8)$$

where $D_{\text{KL}}^{tar}$ represents the target KL-divergence. This approach bounds $\beta_{\text{KL}}$ to be in a reasonable range constrained by $D_{\text{KL}}^{tar}$, leading to robust performance across tasks without requiring manual tuning (4.5).

### 3.4. Supervising Habitual Behaviors with Critic

Habitual behaviors are fast and efficient but should not be purely instinct-driven. In the brain, regions like the dorsal striatum provide feedback to ensure they remain effective and avoid making mistakes (Kang et al., 2021). Inspired by this, Habi also introduces a ***Critic*** function that evaluates habitual decisions by considering both the decision latent

$z$ and the corresponding action $a$ (similar to a Q-function, Figure 3a). The critic loss is defined as:

$$\mathcal{L}_{\text{critic}} = \left\|\text{Critic}(z^q, a^*) - \mathcal{Q}\right\|_2. \quad (9)$$

where $\mathcal{Q}$ is a scalar, represents the decision quality. In offline reinforcement learning, $\mathcal{Q}$ is typically estimated from offline data and represent as a Q-function (Wang et al., 2023; Hansen-Estruch et al., 2023), value function (Hansen-Estruch et al., 2023; Lu et al., 2025), or a classifier (Liang et al., 2023; Janner et al., 2022). In practice, we use the pre-trained $\mathcal{Q}$ corresponding to its diffusion planner as the ground truth of our critic function. Note that $z^q$ is detached (gradient stopped) in critic learning.

### 3.5. Inference with Habitual Behaviors

After habitization training (Figure 3a), we obtained a habitual behavior model, which can be used without planning (Figure 3b). We refer to it as **Habitual Inference** (Figure 3b), which uses the **Prior Encoder**, **Decoder**, and **Critic** to generate habitual behaviors efficiently. HI samples multiple latents $z_i^p$ from the prior distribution, decodes them into corresponding actions $a_i$, and select the best action using the critic. The process is formalized as:

$$\{z_i^p\}_{i=1}^N \sim \mathcal{N}(\mu^p, \sigma^p) \leftarrow \text{PriorEncoder}(s) \quad (10)$$

$$a_i = \text{Decoder}(z_i) \quad (11)$$

$$a = \arg\max_{a_i} \text{Critic}(z_i^p, a_i). \quad (12)$$

where $N$ represents the number of sampling candidates. The Critic evaluates each action $a_i$ along with its decision latent $z_i$, and selects the best action $a$ as the habitual behavior for inference. In practice, we observe using $N = 5$ candidates is sufficient to achieve satisfying performance. Detailed discussions and ablation studies are provided in Section 4.6.

## 4. Experimental Results

### 4.1. Experiment Setup

**Benchmarks.** We empirically evaluate Habi on a diverse set of tasks from the D4RL dataset (Fu et al., 2020), one of the most widely used benchmarks for offline RL. We test our methods across different types of decision-making tasks, including locomotion, manipulation, and navigation (Appendix A).

**Baselines.** To better compare the performance of Habi with other state-of-the-art diffusion planners to benchmark its performance, we include four types of representative baselines: (1) deterministic policies: BC (vanilla imitation learning), SRPO (Chen et al., 2024a) (2) diffusion policies: IDQL (Hansen-Estruch et al., 2023), DQL (Wang et al., 2023), (3) diffusion planners: Diffuser (Janner et al.,

2022), AdaptDiffuser (Liang et al., 2023), Decision Diffuser (DD) (Ajay et al., 2022), Diffusion Veteran (DV) (Lu et al., 2025), and (4) recent works on accelerated decision-making: DiffuserLite (Dong et al., 2024a), DTQL (Chen et al., 2024b). Considering that different baselines adopt varying frameworks and may differ in decision-making frequency, we reproduce the relevant baselines (marked with *) using CleanDiffuser (Dong et al., 2024b), ensuring a fair and consistent comparison.

**Infrastructure** All runtime measurements were conducted on two different computing hardwares: a laptop CPU (Apple M2 Max) or a server GPU (Nvidia A100). Training was on Nvidia A100 GPUs.

**Reproducibility** All the results are calculated over 500 episode seeds for each task to provide a reliable evaluation. HI's results are additionally averaged on 5 training seeds to ensure robustness. Our code is anonymously available at https://bayesbrain.github.io/.

### 4.2. Efficient and Effective Decision Making

How does HI compare with related methods in terms of efficiency and performance? By summarizing the results from the experiments (Table 1), we find that Habitual Inference (HI), could achieve comparable performance of best diffusion policies or diffusion planners. Notably, HI even outperforms the strongest decision-making baselines in certain tasks, suggesting that the habitization process may potentially help mitigate planning errors and enhance decision quality. This phenomenon is particularly evident in tasks requiring precise planning, such as navigation and manipulation tasks.

Besides the standard performance metrics (the average total rewards in an online testing episode), real-world decision-making poses additional requirements on the decision efficiency or frequency. We also conduct a detailed analysis of the decision frequency across different tasks, hardwares, and parallelism levels in Table 4 in the Appendix D.1. HI demonstrates a significant decision speed margin over other baselines. Figure 4 illustrates the trade-off between performance and frequency across different tasks, highlighting HI's ability to balance efficiency and effectiveness.

### 4.3. Comparison with Direct Distillation Methods

How much performance gain can we obtain using Habi **comparing with direct distillation** of diffusion planner? We compare it with standard distillation methods. Specifically, we evaluate (1) *HI w/o Critic*, which removes the critic and directly uses the habitual policy without selection (Figure 3b), and (2) *Standard Distillation*, which applies supervised learning to mimic the planner. The results in Table 2 show that even without the critic, Habi consistently

*Table 1.* **Performance comparison on the D4RL benchmark.** The reported values are Mean $\pm$ Standard Error over 500 episode seeds for robust testing. Frequencies are measured using a CPU (Apple M2 Max, MacBook laptop). Baselines marked with * were reproduced using CleanDiffuser (Dong et al., 2024b) for consistency. Deterministic policies such as BC, SRPO (Chen et al., 2024a) are theoretically more efficient, however, their performance remains substantially inferior to probabilistic policies. The best results are highlighted in **bold**, while the second-best results are underlined. Action frequency on GPU can be found in Table 4.

| Tasks | | Deterministic Policies | | Diffusion Policies | | Diffusion Planners | | | | Accelerated Probabilistic Decision-Making | | | |
|---|---|---|---|---|---|---|---|---|---|---|---|---|---|
| Dataset | Environment | BC | SRPO | IDQL* | DQL* | Diffuser | AdaptDiffuser | DD | DV | DL-R1* | DL-R2* | DTQL | HI (Ours) |
| Medium-Expert | HalfCheetah | 35.8 | 92.2 | 91.3 ± 0.6 | 96.0 ± 0.0 | 88.9 ± 0.3 | 89.6 ± 0.8 | 90.6 ± 1.3 | 92.7 ± 0.3 | 90.6 ± 0.7 | 88.6 ± 0.7 | 92.7 ± 0.2 | 98.0 ± 0.0 |
| Medium-Replay | HalfCheetah | 38.4 | 51.4 | 46.5 ± 0.3 | 47.8 ± 0.0 | 37.7 ± 0.5 | 38.3 ± 0.9 | 39.3 ± 4.1 | 45.8 ± 0.1 | 44.0 ± 0.2 | 41.8 ± 0.2 | 50.9 ± 0.1 | 48.5 ± 0.0 |
| Medium | HalfCheetah | 36.1 | 60.4 | 51.5 ± 0.1 | 52.3 ± 0.0 | 42.8 ± 0.3 | 44.2 ± 0.6 | 49.1 ± 1.0 | 50.4 ± 0.0 | 46.9 ± 0.1 | 45.9 ± 0.2 | 57.9 ± 0.1 | 53.5 ± 0.0 |
| Medium-Expert | Hopper | 111.9 | 101.1 | 110.1 ± 0.7 | 111.4 ± 0.2 | 103.3 ± 1.3 | 111.6 ± 2.0 | 111.8 ± 1.8 | 110.0 ± 0.5 | 111.1 ± 0.2 | 110.9 ± 0.4 | 109.3 ± 1.5 | 92.4 ± 2.0 |
| Medium-Replay | Hopper | 11.3 | 101.2 | 99.4 ± 0.1 | 102.2 ± 0.0 | 93.6 ± 0.4 | 92.2 ± 1.5 | 100.0 ± 0.7 | 91.9 ± 0.0 | 89.7 ± 0.4 | 94.7 ± 0.4 | 100.0 ± 0.1 | 102.0 ± 0.0 |
| Medium | Hopper | 29.0 | 95.5 | 70.1 ± 2.0 | 102.3 ± 0.0 | 74.3 ± 1.4 | 96.6 ± 2.7 | 79.3 ± 3.6 | 83.6 ± 1.2 | 97.4 ± 0.5 | 95.9 ± 1.0 | 99.6 ± 0.9 | 102.5 ± 0.1 |
| Medium-Expert | Walker2d | 6.4 | 114 | 110.6 ± 0.0 | 111.7 ± 0.0 | 106.9 ± 0.2 | 108.2 ± 0.8 | 108.8 ± 1.7 | 109.2 ± 0.0 | 109.2 ± 0.7 | 108.1 ± 1.0 | 110.0 ± 0.1 | 113.0 ± 0.0 |
| Medium-Replay | Walker2d | 11.8 | 84.6 | 89.1 ± 2.4 | 101.2 ± 0.3 | 70.6 ± 1.6 | 84.7 ± 3.1 | 75.0 ± 4.3 | 85.0 ± 0.5 | 85.0 ± 0.6 | 84.1 ± 0.5 | 88.5 ± 2.2 | 102.0 ± 0.0 |
| Medium | Walker2d | 6.6 | 84.4 | 88.1 ± 0.4 | 90.0 ± 0.5 | 79.6 ± 0.6 | 84.4 ± 2.6 | 82.5 ± 1.4 | 82.8 ± 0.1 | 82.3 ± 0.7 | 83.9 ± 0.7 | 89.4 ± 0.1 | 91.3 ± 0.1 |
| **Performance** | | 31.9 | 87.2 | 84.1 | **90.5** | 77.5 | 83.3 | 81.8 | 83.5 | 84.0 | 83.8 | 88.7 | 89.2 |
| **Action Frequency on CPU (Hz)** | | – | – | 35.9 | 197.2 | 0.23 | 0.23 | 16.7 | 7.5 | 75.5 | 206.6 | 142.7 | **1329.7** |
| Mixed | Kitchen | 47.5 | – | 66.5 ± 4.1 | 55.1 ± 1.58 | 52.5 ± 2.5 | 51.8 ± 0.8 | 75.0 ± 0.0 | 73.6 ± 0.1 | 60.5 ± 1.0 | 52.6 ± 1.0 | 60.2 ± 0.59 | 69.8 ± 0.4 |
| Partial | Kitchen | 33.8 | – | 66.7 ± 2.5 | 65.5 ± 1.38 | 55.7 ± 1.3 | 55.5 ± 0.4 | 56.5 ± 5.8 | 94.0 ± 0.3 | 43.5 ± 1.6 | 37.8 ± 1.8 | 74.4 ± 0.25 | 94.8 ± 0.6 |
| **Performance** | | 40.7 | – | 66.6 | 60.3 | 54.1 | 53.7 | 65.8 | **83.8** | 52.0 | 45.2 | 67.3 | 82.3 |
| **Action Frequency on CPU (Hz)** | | – | – | 34.4 | 146.0 | 0.05 | 0.06 | 27.1 | 2.7 | 16.5 | 36.2 | 97.5 | **385.7** |
| Diverse | Antmaze-Large | 0.0 | 53.6 | 40.0 ± 11.4 | 70.6 ± 3.7 | 27.3 ± 2.4 | 8.7 ± 2.5 | 0.0 ± 0.0 | 80.0 ± 1.8 | 0.0 ± 0.0 | 32.6 ± 3.8 | 54.0 ± 2.2 | 65.2 ± 2.0 |
| Play | Antmaze-Large | 0.0 | 53.6 | 48.7 ± 4.7 | 81.3 ± 3.1 | 17.3 ± 1.9 | 5.3 ± 3.4 | 0.0 ± 0.0 | 76.4 ± 2.0 | 54.0 ± 4.0 | 71.3 ± 3.6 | 52.0 ± 2.2 | 81.7 ± 1.7 |
| Diverse | Antmaze-Medium | 0.0 | 75.0 | 83.3 ± 5.0 | 82.6 ± 3.0 | 2.0 ± 1.6 | 6.0 ± 3.3 | 4.0 ± 2.8 | 87.4 ± 1.6 | 85.33 ± 2.8 | 86.6 ± 2.7 | 82.2 ± 1.7 | 88.8 ± 1.4 |
| Play | Antmaze-Medium | 0.0 | 80.7 | 67.3 ± 5.7 | 87.3 ± 2.7 | 6.7 ± 5.7 | 12.0 ± 7.5 | 8.0 ± 4.3 | 89.0 ± 1.6 | 79.3 ± 3.3 | 78.0 ± 3.3 | 79.6 ± 1.8 | 85.3 ± 1.5 |
| **Performance** | | 0.0 | 65.7 | 59.8 | 80.5 | 13.3 | 8.0 | 3.0 | 83.2 | 54.7 | 67.1 | 67.0 | 80.3 |
| **Action Frequency on CPU (Hz)** | | – | – | 34.2 | 198.6 | 0.03 | 0.03 | 11.8 | 0.7 | 15.7 | 35.2 | 138.6 | **908.3** |
| Large | Maze2D | 5.0 | – | 167.4 ± 5.3 | 186.8 ± 1.7 | 123 | 167.9 ± 5.0 | – | 203.6 ± 1.4 | 103.3 ± 7.2 | 50.1 ± 6.8 | – | 199.2 ± 2.0 |
| Medium | Maze2D | 30.3 | – | 133.9 ± 3.0 | 152.0 ± 0.8 | 121.5 | 129.9 ± 4.6 | – | 150.7 ± 1.0 | 52.1 ± 5.6 | 106.9 ± 7.4 | – | 150.1 ± 1.5 |
| Umaze | Maze2D | 3.8 | – | 119.6 ± 4.1 | 140.6 ± 1.0 | 113.9 | 135.1 ± 5.8 | – | 136.6 ± 1.3 | 36.4 ± 7.0 | 125.0 ± 6.9 | – | 144.3 ± 1.7 |
| **Performance** | | 13.0 | – | 140.3 | 159.8 | 119.5 | 144.3 | – | 163.6 | 63.9 | 94.0 | – | **164.5** |
| **Action Frequency on CPU (Hz)** | | – | – | 33.9 | 215.8 | 0.03 | 0.03 | – | 2.8 | 16.4 | 38.5 | – | **1532.6** |

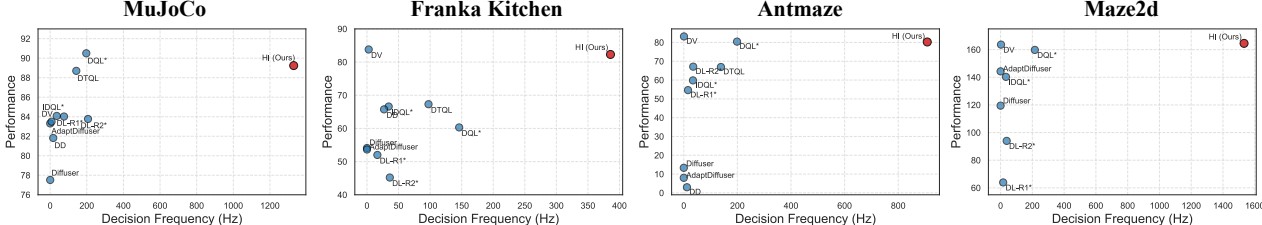

*Figure 4.* Visualized results of Table 1. HI consistently performs in parallel with best models while being highly efficient.

outperforms standard distillation across all tasks.

This demonstrates that HI itself contributes to effective decision-making beyond simple imitation. While the critic further refines decision quality, the strong performance of *HI w/o Critic* confirms that Habi is not merely replicating planner behavior but instead leveraging the habitization process to learn an efficient and robust decision policy.

### 4.4. Visualizing Prior and Posterior Policy Distributions

To investigate whether Habi can reasonably **align habitual and goal-directed decision spaces** we conduct a case study by visualing the action distributions of the Habitual Inference (HI) policy and its corresponding diffusion-based planner in the Maze2D environment, where the two-dimensional action space allows for a clear comparison. Figure 5 visualizes the action distributions of Diffusion Veteran (DV) (Lu et al., 2025), a state-of-the-art diffusion planner, and its corresponding habitual inference policy generated using our framework Habi.

We find that despite the fundamental differences in inference mechanisms, the action distributions of HI remain well-aligned with those of DV across different states. The action distributions exhibit strong structural consistency, with HI capturing both the variability and intent of the planner's actions. Notably, HI produces more concentrated distributions. This may reflect its ability to commit to high-confidence decisions without iterative refinement.

### 4.5. Robustness of Adaptive KL-Divergence Weighting

We used adaptive $\beta_{\mathrm{KL}}$ (Equation 8) in habitization learning for all tasks. Then a natural question is: **how about manually tuning $\beta_{\mathrm{KL}}$ on each task?** We evaluate the effectiveness of our proposed adaptive $\beta_{\mathrm{KL}}$. As shown in Figure 6, fixed $\beta_{\mathrm{KL}}$ tuning requires extensive grid search, and its performance is highly sensitive to the choice of $\beta_{\mathrm{KL}}$. A suboptimal $\beta_{\mathrm{KL}}$ can lead to either poor alignment (when too small) or degraded reconstruction quality (when too large). This issue is especially pronounced in complex tasks like AntMaze and Franka Kitchen, where improper

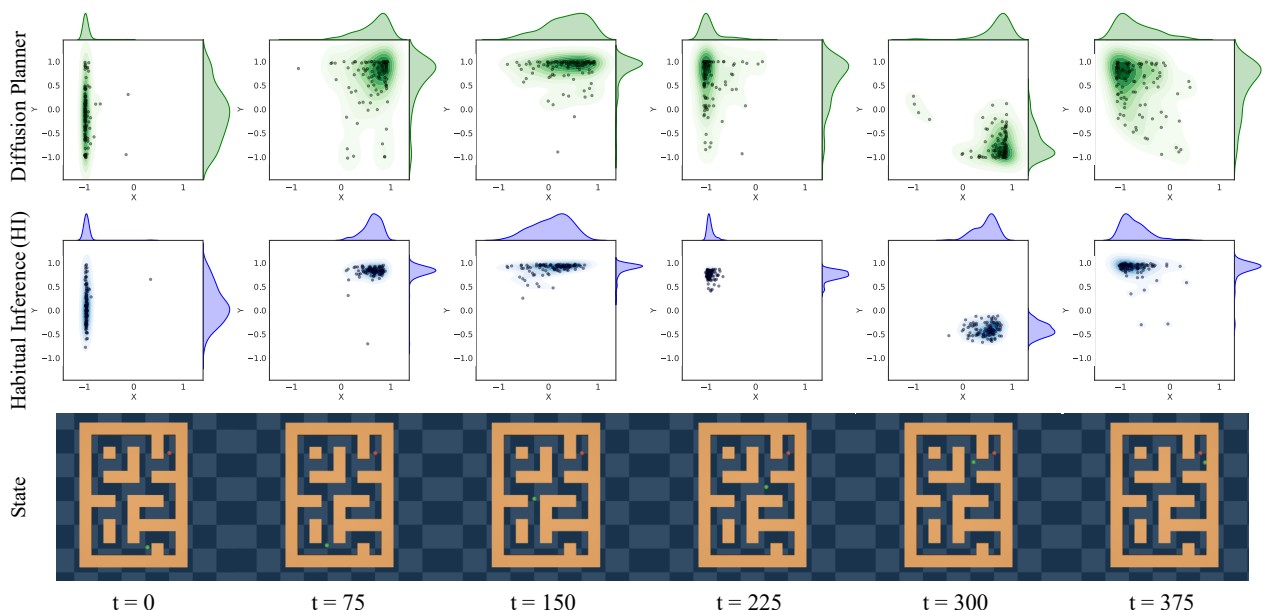

*Figure 5.* **Action distributions of Diffusion Planner (DV) and Habitual Inference (HI).** Visualization of the action distributions from a state-of-the-art diffusion planner (DV, 2.8Hz, top) (Lu et al., 2025) and its corresponding Habitual Inference policy (HI, 1532.6Hz, middle) generated by our Habi framework. Here, HI shows a probabilistic generation capacity while roughly aligning with the action distribution from the diffusion planner (DV). More examples are deferred to Appendix D.4.

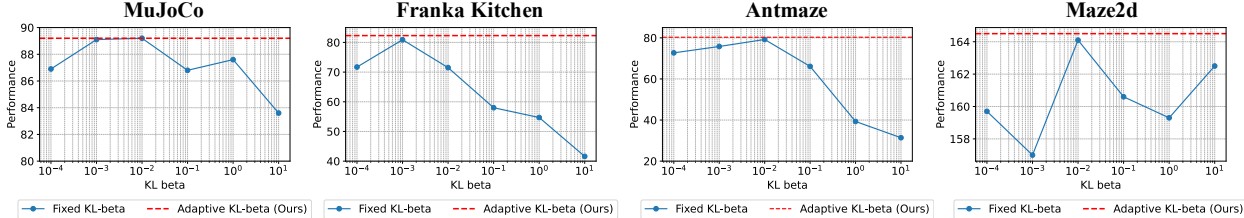

*Figure 6.* **Performance Comparison of Searching $\beta_{\mathrm{KL}}$ and our Adaptive $\beta_{\mathrm{KL}}$ Mechanism.** Fixed $\beta_{\mathrm{KL}}$ requires grid search for each task to achieve peak performance, while our adaptive $\beta_{\mathrm{KL}}$ achieves comparable performance **without task-specific tuning**, as shown by the consistent red dashed line across all tasks.

weighting significantly degrades performance.

In contrast, our adaptive $\beta_{\mathrm{KL}}$ mechanism consistently achieves almost optimal performance across all tasks without requiring task-specific tuning. By dynamically adjusting $\beta_{\mathrm{KL}}$ during training, it effectively balances decision space alignment and reconstruction accuracy in an automated manner. This not only simplifies training but also ensures robustness across diverse environments. The consistent performance of adaptive $\beta_{\mathrm{KL}}$ (red dashed line) highlights its reliability in achieving strong results without extensive hyperparameter tuning, making Habi potentially more scalable and adaptable across diverse tasks with minimal human intervention.

### 4.6. Importance of Action Selection with Sampling

We introduced action selection with sampling in HI (Figure 3b), where more numbers of candidates will bring more computational cost. Here we examine the **necessity of this design choice and the proper number of candidates**. As

shown in Figure 7, selecting from multiple candidates consistently improves decision quality across various tasks, confirming the benefit of leveraging the Critic for filtering. Interestingly, we observe that using a single candidate ($N = 1$) already achieves competitive results in some environments, demonstrating that Habi can function effectively even in an ultra-lightweight setting. However, without candidate selection, the policy loses the ability to refine habitual decisions, leading to suboptimal outcomes in more complex tasks like AntMaze and Maze2D. A moderate number of candidates (e.g., $N = 5$) provides a good balance, achieving near-optimal performance with minimal computational overhead.

## 5. Related work

**Accelerating Diffusion Decision Making** Diffusion models have been widely applied to decision-making tasks, including offline RL, online RL, and imitation learning. However, their slow inference speed remains a significant bot-

*Table 2.* **Comparison with alternative approaches.** We compare **HI** with (1) HI without critic-based selection; and (2) standard distillation, which directly applies supervised learning to distill the corresponding diffusion planner.

| Tasks | | Methods | | |
|---|---|---|---|---|
| **Dataset** | **Environment** | **HI (Ours)** | HI w/o Critic | Direct Distill |
| Med-Exp | HalfCheetah | 98.0 ± 0.0 | 96.9 ± 0.1 | 97.1 ± 0.1 |
| Med-Rep | HalfCheetah | 48.5 ± 0.0 | 47.0 ± 0.0 | 46.7 ± 0.1 |
| Medium | HalfCheetah | 53.5 ± 0.0 | 51.2 ± 0.0 | 51.2 ± 0.1 |
| Med-Exp | Hopper | 92.4 ± 2.0 | 85.9 ± 2.0 | 61.1 ± 1.5 |
| Med-Rep | Hopper | 102.0 ± 0.0 | 101.8 ± 0.0 | 101.4 ± 0.2 |
| Medium | Hopper | 102.5 ± 0.1 | 98.5 ± 1.1 | 102.2 ± 0.0 |
| Med-Exp | Walker2d | 113.0 ± 0.0 | 112.5 ± 0.0 | 112.2 ± 0.0 |
| Med-Rep | Walker2d | 102.0 ± 0.0 | 101.9 ± 0.2 | 101.0 ± 0.0 |
| Medium | Walker2d | 91.3 ± 0.1 | 91.4 ± 0.3 | 91.9 ± 0.1 |
| **Performance** | | **89.2** | 87.5 (-2.0%) | 85.0 (-4.9%) |
| Mixed | Kitchen | 69.8 ± 0.4 | 66.6 ± 0.4 | 69.1 ± 0.4 |
| Partial | Kitchen | 94.8 ± 0.6 | 94.2 ± 0.6 | 64.8 ± 0.8 |
| **Performance** | | **82.3** | 80.4 (-2.3%) | 67.0 (-19.1%) |
| Diverse | Antmaze-Large | 65.2 ± 2.0 | 3.8 ± 0.8 | 72.0 ± 1.6 |
| Play | Antmaze-Large | 81.7 ± 1.7 | 77.8 ± 1.8 | 52.0 ± 1.6 |
| Diverse | Antmaze-Medium | 88.8 ± 1.4 | 92.1 ± 1.1 | 44.0 ± 1.5 |
| Play | Antmaze-Medium | 85.3 ± 1.5 | 88.0 ± 1.4 | 76.0 ± 1.2 |
| **Performance** | | **80.3** | 65.4 (-18.5%) | 61.0 (-29.4%) |
| Large | Maze2D | 199.2 ± 2.0 | 201.8 ± 1.9 | 198.9 ± 1.8 |
| Medium | Maze2D | 150.1 ± 1.5 | 143.9 ± 1.6 | 143.0 ± 1.7 |
| Umaze | Maze2D | 144.3 ± 1.7 | 138.3 ± 1.8 | 137.7 ± 1.8 |
| **Performance** | | **164.5** | 161.3 (-1.9%) | 159.9 (-2.9%) |

tleneck, limiting their broader applicability in real-world scenarios. A primary approach to mitigating this issue involves specialized diffusion solvers (Song et al., 2020; Lu et al., 2022) to reduce sampling steps, while varying degrees of performance degradation remain unavoidable. More recently, distillation-based approaches (Poole et al., 2023; Wang et al., 2024b) have been proposed to enable one-step generation in diffusion models. However, directly applying these techniques to decision-making may lead to suboptimal performance (Chen et al., 2024b;a).

To achieve both effective and efficient decision-making, ODE reflow-based numerical distillation has been introduced as a promising direction (Dong et al., 2024a). Meanwhile, recent studies also explore joint training of a diffusion model and a single-step policy, using the latter for deployment while maintaining state-of-the-art performance (Chen et al., 2024b). These works serve as important baselines for our comparison.

**Variational Bayes in RL** Variational Bayesian (VB) methods have been incorporated into RL research for several purposes. A popular usage is exploiting the generative capacity of VAE and its recurrent versions (Chung et al., 2015) for constructing the world model (Ha & Schmidhuber, 2018; Hafner et al., 2018; 2023). VB methods are also used to extract useful representation of environmental states from raw observations by modeling the state transitions (Igl et al., 2018; Han et al., 2020; Ni et al., 2024). The acquired representation is then used for the original RL task, i.e. maximizing rewards. Also, VB principles can help to design

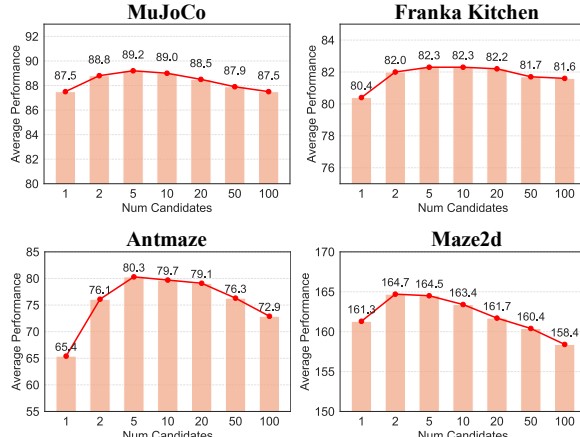

*Figure 7.* **Effect of the number of sampling candidates on performance.** Increasing the number of candidates $N$ improves performance initially, but excessive candidates ($N \geq 50$) lead to diminishing returns or slight degradation. A moderate choice (e.g., $N = 5$) provides a good balance between performance and efficiency. Note that $N = 1$ is the case without critic-based selection.

RL objective functions (Levine, 2018; Fellows et al., 2019; Guan et al., 2024). Other usages include using the prediction error to compute curiosity-driven reward to encourage exploration (Yin et al., 2021). Moreover, Han et al. (2022; 2024) demonstrated that ELBO can connect two policies in online RL, with the one easier to learn as posterior, and the prior could be used for inference, which inspires our study. However, Habi focuses on offline RL, and we are the first to show such an ELBO can result in tremendous speedup without significant performance degradation.

## 6. Conclusion

In this work, we introduce **Habi**, a general framework that habitizes diffusion planning, transforming slow, goal-directed decision-making processes into efficient, habitual behaviors. Through comprehensive evaluations on standard offline RL benchmarks, we demonstrate that Habi achieves orders-of-magnitude speedup while maintaining or even surpassing the performance of state-of-the-art diffusion-based decision-making approaches.

Looking forward, Habi demonstrates potential for real-time, high-speed decision-making in real-world environments, making it a promising approach for applications in embodied AI and other real-world decision-making problems. Nevertheless, our work has limitations. We have focused on state (proprioception)-based decision-making tasks using offline datasets. Future directions include extending Habi to online RL, investigating its generalization to broader domains such as vision-based tasks (Du et al., 2024; Chi et al., 2023), examining its effectiveness in real-world tasks, as well as combining with orthogonal techniques such as ensemble approaches (An et al., 2021; Yang et al., 2022).

## Impact Statement

This work contributes to technical advancements in machine learning by proposing an efficient decision-making framework. It does not introduce unique ethical concerns beyond standard considerations in algorithmic development and deployment.

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

## A. Benchmarking Tasks

As shown in Figure 8, we consider a diverse set of benchmarking tasks to evaluate the performance of Habi. These tasks are designed to encompass a wide range of evaluation metrics, including locomotion tasks that emphasize short-term planning, robotic arm tasks requiring long-term strategic planning, and navigation tasks focused on path planning. We provide a detailed description of each task below.

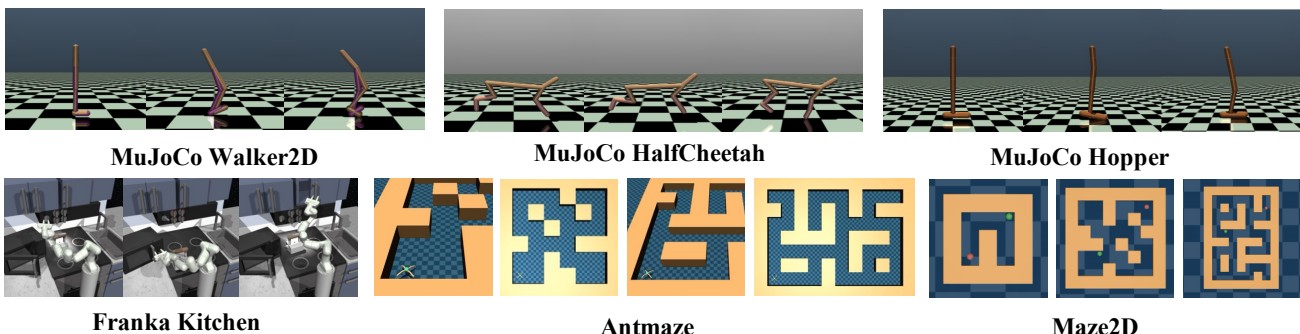

**MuJoCo Walker2D**  **MuJoCo HalfCheetah**  **MuJoCo Hopper**

**Franka Kitchen**  **Antmaze**  **Maze2D**

*Figure 8.* **Rendering of the benchmarking tasks considered in this study.** The tasks encompass a diverse set of evaluation metrics, including locomotion tasks that emphasize short-term planning, robotic arm tasks requiring long-term strategic planning, and navigation tasks focused on path planning.

**MuJoCo Locomotion**  The MuJoCo Locomotion task is a standard benchmarking task in reinforcement learning that requires the agent to control a simulated robot to navigate through a complex environment. The task is designed to evaluate the agent's ability to perform locomotion tasks that require short-term planning.

**Franka Kitchen**  The Kitchen task is a robotic arm manipulation task that requires the agent to manipulate objects in a kitchen environment. The task is designed to evaluate the agent's ability to perform long-term strategic planning and manipulation tasks.

**AntMaze**  The AntMaze task is a combination of locomotion and planning tasks that require the agent to navigate through a maze environment. The variations of this environment can be initialized with different maze configurations and increasing levels of complexity. The task is designed to evaluate the agent's ability to perform locomotion tasks while incorporating planning strategies.

**Maze2D**  The Maze2D task is a pure navigation task that requires the agent to navigate through a 2D maze environment. The variations of this environment can be initialized with different maze configurations and increasing levels of complexity. These tasks are used to test planning capabilities in environments where spatial reasoning is critical.

## B. Property of ELBO

While the ELBO (Equation 1) optimizes the log-likelihood of reconstruction data, $\mathbb{E}_{z \sim q(z)} \log P(x(z))$, based on the exception over posterior distribution of $z$, the likelihood over the expectation of prior distribution, $\mathbb{E}_{z \sim p(z)} P(x(z))$, is also being optimized. Equivalently, we need to show that the ELBO is indeed a lower bound on $\log \mathbb{E}_{z \sim p(z)} P(x|z)$:

$$\mathbb{E}_{z \sim q(z)} \log P(x|z) - D_{\mathrm{KL}}[q(z)||p(z)] \leq \log \mathbb{E}_{z \sim p(z)} P(x|z). \tag{13}$$

We start with the definition of the marginal likelihood:

$$\log \mathbb{E}_{z \sim p(z)} P(x) = \log \int p(x, z) dz = \log \int p(x|z) p(z) dz. \tag{14}$$

Introducing a variational distribution $q(z)$, we rewrite this using importance sampling:

$$\log \mathbb{E}_{z \sim p(z)} P(x) = \log \int q(z) \frac{p(x|z) p(z)}{q(z)} dz. \tag{15}$$

Applying Jensen's inequality (since $\log$ is concave), we obtain:

$$\log \mathbb{E}_{z \sim p(z)} P(x) \geq \int q(z) \log \frac{p(x|z)p(z)}{q(z)} dz. \tag{16}$$

Rewriting the right-hand side:

$$\mathbb{E}_{z \sim q(z)} \log p(x|z) + \mathbb{E}_{z \sim q(z)} \log \frac{p(z)}{q(z)}. \tag{17}$$

Recognizing the second term as the negative Kullback-Leibler (KL) divergence:

$$\mathbb{E}_{z \sim q(z)} \log p(x|z) - D_{\text{KL}}(q(z)||p(z)). \tag{18}$$

Since the KL divergence is always non-negative, we conclude that:

$$\mathbb{E}_{z \sim q(z)} \log p(x|z) - D_{\text{KL}}(q(z)||p(z)) \leq \log \mathbb{E}_{z \sim p(z)} P(x). \tag{19}$$

Thus, the ELBO is a valid lower bound on $\log P(x)$. This ensures that our ELBO-like objective function (Equation 7) improves habitual (prior) policy.

## C. Implementation Details

In this section, we provide additional implementation details of Habi, including the network architecture and hyperparameters used in all experiments.

### C.1. Network Architecture

As demonstrated in Figure 3, the Habi pipeline consists of four main components: (1) *Prior Encoder*, (2) *Posterior Encoder*, (3) *Decoder*, and (4) *Critic*. All these modules can be implemented with a lightweight multiple-layer perceptrons (MLP) and trained using standard backpropagation during the Habitization stage. We provide detailed descriptions of each component below.

**Prior Encoder**  The Prior Encoder is required to encode the state into the prior latent $z^p$ for generating habitual behaviors. In Habi, it is implemented with a multiple-layer perceptron (MLP), that only takes the current state $s$ as input and outputs the mean $\mu^p$ and standard deviation $\sigma^p$ of the prior latent $z^p \sim \mathcal{N}(\mu^p, \sigma)$:

$$\begin{aligned} \mu^p &= \text{MLP}(s_t) \\ \xi^p &= \text{MLP}(s_t) \\ \sigma^p &= \text{softplus}(\xi^p) + \epsilon, \end{aligned} \tag{20}$$

where $\epsilon$ is a small constant to ensure numerical stability ($\epsilon = 0.01$ in our work).

**Posterior Encoder**  The Posterior Encoder is responsible for encoding the state-action pair $(s, a)$ into the posterior latent $z^q$ for generating goal-directed behaviors. It is implemented similarly to the Prior Encoder, with the only difference being that it takes the state-action pair as input:

$$\begin{aligned} \mu^q &= \text{MLP}(s_t, a^*) \\ \xi^q &= \text{MLP}(s_t, a^*) \\ \sigma^q &= \text{softplus}(\xi^q) + \epsilon. \end{aligned} \tag{21}$$

**Decoder**  The Decoder is responsible for decoding the decision latent $z$ into an action $a$. In Habi, the Decoder is implemented as a simple MLP that takes the latent $z$ as input and outputs the action $a$:

$$a = \text{MLP}(z). \tag{22}$$

**Critic**   The Critic is used to evaluate the quality of habitual behaviors generated by the Decoder. In Habi, the Critic is implemented as a simple MLP that takes the decision latent $z$ and the action $a$ as input and outputs a scalar value $\mathcal{Q}$ representing the decision quality:

$$\mathcal{Q} = \text{MLP}(z, a). \tag{23}$$

In all experiments of this work, these components are consistently implemneted as a lightweight three-layer MLP with hidden dimension of 256, as is shown in Table 3. The Direct Distill baseline uses *the same* architecture as Habi but performs imitation learning directly on state-action pairs provided by diffusion planners.

## C.2. Full Hyperparameters

We provide the consistent hyperparameters used in all experiments in Table 3.

*Table 3.* **Hyperparameters in our experiments**.

| Settings | Value |
|---|---|
| Optimizer | Adam |
| Learning Rate | 3e-4 |
| Gradient Steps | 1000000 |
| Batch Size | 256 |
| Latent Dimension: $\text{Dim}(z)$ | 256 |
| MLP Hidden Size (Encoder & Decoder) | 256 |
| MLP Hidden Layers (Encoder & Decoder) | 2 |
| Habitization Target (Locomotion Related) | DQL (Wang et al., 2023) (MuJoCo, Antmaze) |
| Habitization Target (Planning Related) | DV (Lu et al., 2025) (Kitchen, Maze2D) |
| Target KL-divergence $D_{\text{KL}}^{tar}$ | 1.0 |
| Number of Sampling Candidates in Habitization training | 50 |
| Number of Sampling Candidates in Habitual Inference | 5 |

## C.3. Policy and Critic Training Details

To provide further details on the loss functions introduced in Section 3.3 and Section 3.4, we elaborate on the reconstruction loss (Equation (4)) and the critic training loss (Equation (9)) in this section.

**Policy Training.**   Most state-of-the-art diffusion model-based policy models generate actions by **sampling a batch of candidate actions** and selecting the best one for decision-searching, recent studies (Wang et al., 2023; Dong et al., 2024b; Lu et al., 2025) also have validated its effectiveness. In Habi, we aim to learn a policy that directly habitize their final decisions. Specifically, the reconstruction loss optimizes the decoder to recover the planner's selected action:

$$\mathcal{L}_{\text{recon}} = \left\| \text{Decoder}(z^q) - a^* \right\|_2, \tag{24}$$

where $a^*$ is the best action chosen by the planner from the candidate batch. This helps the habitual policy focus on habitizing high-quality decision-making rather than reproducing the full distribution of candidate actions, ensuring efficient and reliable behavior generation.

**Critic Training.**   Unlike the policy, which only needs to learn the planner's final decision, the critic benefits from exposure to both optimal and suboptimal plans. A robust critic should distinguish between high- and low-quality decisions, enabling more effective filtering of habitual actions during inference. To achieve this, we train the critic using the whole batch of candidate actions. Formally, the critic loss is given by:

$$\mathcal{L}_{\text{critic}} = \frac{1}{N} \sum_{i}^{N} \left\| \text{Critic}(z^q, a_i^*) - \mathcal{Q}_i \right\|_2, \tag{25}$$

where $a_i^*$ also includes suboptimal actions that were not selected by the planner, and $Q_i$ is the corresponding value given by the pretrained value function of the diffusion planners (Wang et al., 2023; Lu et al., 2025). This exposure to diverse plans allows the critic to generalize better and maintain robustness in habitual decision-making.

# D. Extensive Experimental Results

## D.1. Decision Frequency Evaluation across Different Devices

We evaluate the decision frequency of our methods across tasks, devices, and levels of parallelism (Table 4). On a PC laptop (Apple M2 Max CPU), HI consistently achieves the highest decision frequency, significantly outperforming diffusion policies (IDQL, DQL) and diffusion planners (Diffuser, AdaptDiffuser, DD, DV). Compared with other accelerated probabilistic decison-making methods, HI demonstrates a consistent advantage in decision frequency across all tasks.

On a server with an Nvidia A100 GPU, HI maintains its advantage of high decision frequency. At 10 and 20 parallel environments, it leads in Franka Kitchen, Antmaze, and Maze2D. In MuJoCo with 10 environments, DTQL achieves a slightly higher frequency, but HI remains competitive while being significantly faster than diffusion-based methods.

The results demonstrate that HI is capable of making decisions at a high frequency across different tasks, devices, and levels of parallelism, making it a suitable choice for real-time decision-making applications.

*Table 4.* **Decision-making frequency across tasks, devices and parallelism levels.** Frequency reflects the number of actions (or action batches) generated per second using either a PC laptop (CPU@Apple M2 Max) or a Linux server (GPU@Nvidia A100) under varying levels of parallelism. The best results are highlighted in **bold**.

| Task | Device | Parallel Envs | IDQL* | DQL* | Diffuser | AdaptDiffuser | DD | DV | DL-R1* | DL-R2* | DTQL | HI (Ours) |
|------|--------|---------------|-------|------|----------|---------------|-----|-----|--------|--------|------|-----------|
| | PC (CPU) | 1 | 35.9 | 197.2 | 0.2 | 0.2 | 16.7 | 7.5 | 75.5 | 206.6 | 142.7 | **1329.7** |
| MuJoCo | Server (GPU) | 10 | 110.9 | 135.4 | 3.1 | 3.1 | 22.9 | 18.0 | 49.4 | 109.2 | **318.8** | 308.4 |
| | Server (GPU) | 20 | 86.5 | 108.1 | 3.0 | 3.0 | 22.4 | 19.8 | 48.4 | 104.2 | 223.7 | **281.3** |
| | PC (CPU) | 1 | 34.4 | 146.0 | 0.1 | 0.1 | 27.1 | 2.7 | 16.5 | 36.2 | 97.5 | **385.7** |
| Franka Kitchen | Server (GPU) | 10 | 78.4 | 92.9 | 2.3 | 2.4 | 22.1 | 7.8 | 27.2 | 51.0 | 143.6 | **147.3** |
| | Server (GPU) | 20 | 64.2 | 85.2 | 2.2 | 2.2 | 21.5 | 4.2 | 24.0 | 36.3 | 113.0 | **137.3** |
| | PC (CPU) | 1 | 34.2 | 198.6 | 0.03 | 0.03 | 11.8 | 0.7 | 15.7 | 35.2 | 138.6 | **908.3** |
| Antmaze | Server (GPU) | 10 | 102.3 | 127.6 | 2.5 | 2.3 | 24.3 | 2.0 | 29.8 | 58.0 | 241.4 | **261.4** |
| | Server (GPU) | 20 | 80.1 | 100.1 | 1.5 | 1.5 | 24.2 | 1.0 | 24.9 | 50.8 | 196.3 | **251.5** |
| | PC (CPU) | 1 | 33.9 | 215.8 | 0.03 | 0.03 | – | 2.8 | 16.4 | 38.5 | – | **1532.6** |
| Maze2D | Server (GPU) | 10 | 126.4 | 151.9 | 2.7 | 2.6 | – | 8.3 | 30.3 | 62.5 | – | **388.4** |
| | Server (GPU) | 20 | 90.8 | 140.8 | 1.5 | 1.5 | – | 4.4 | 24.8 | 52.6 | – | **376.0** |

## D.2. Impact of Number of Candidates

To further analyze the role of candidate selection, we provide additional results in Table 6, extending our discussion from the main text. The results confirm that selecting from multiple candidates consistently improves decision quality across various tasks. Even with a single candidate ($N = 1$), HI remains competitive, reinforcing its efficiency in lightweight settings. However, increasing $N$ enables further refinement, particularly benefiting complex tasks such as AntMaze and Maze2D, where decision quality is more sensitive to suboptimal habitual actions.

We also observe diminishing returns beyond a moderate number of candidates. While larger $N$ improves performance, gains plateau after $N = 5$ in most environments, suggesting that a small candidate pool is sufficient for near-optimal decision-making. This balance allows HI to achieve high efficiency without excessive computational overhead, making it adaptable across different settings.

## D.3. Additional comparison results

We also provide additional comparisons with accelerated generative decision-making methods, including flow-based (Zhang et al., 2024; Dong et al., 2024a), consistency-based methods (Chen et al., 2023b) on shared tasks according to the original papers. As shown in Table 5, our method (HI) achieves consistently strong performance across a range of MuJoCo environments. These results highlight the effectiveness of our approach in balancing planning quality and efficiency across diverse offline settings. However, we do not want to convey the message that Habi is better than flow-based method. Indeed, they are orthogonal and can be combined (Kingma et al., 2016) for future work.

*Table 5.* Additional comparison results on tasks of MuJoCo

| Environment | HI (Ours) | FTB | CPIQL | CPQL |
|---|---|---|---|---|
| HalfCheetah-ME | 98.0 ± 0.0 | 85.2 ± 0.7 | 81.0 ± 1.7 | 97.8 ± 0.5 |
| HalfCheetah-MR | 48.5 ± 0.0 | 38.4 ± 1.3 | 48.0 ± 1.4 | 46.6 ± 0.8 |
| HalfCheetah-M | 53.5 ± 0.0 | - | 54.6 ± 1.0 | 56.9 ± 0.9 |
| Hopper-ME | 92.4 ± 2.0 | 111.1 ± 2.0 | 110.6 ± 1.4 | 110.4 ± 3.2 |
| Hopper-MR | 102.0 ± 0.0 | 89.6 ± 4.9 | 100.6 ± 1.5 | 97.7 ± 4.6 |
| Hopper-M | 102.5 ± 0.1 | - | 99.7 ± 2.0 | 99.9 ± 4.5 |
| Walker-ME | 113.0 ± 0.0 | 109.3 ± 0.3 | 110.9 ± 0.2 | 110.9 ± 0.1 |
| Walker-MR | 102.0 ± 0.0 | 79.1 ± 1.4 | 91.8 ± 2.8 | 93.6 ± 5.6 |
| Walker-M | 91.3 ± 0.1 | - | 86.2 ± 0.6 | 82.1 ± 2.4 |
| Tasks Average | 89.2 | - | 87.0 | 88.4 |
| Shared Tasks Average | 92.7 | 85.5 | 90.5 | 92.8 |

*Table 6.* **Sweeping Number of Sampling Candidates.** This table presents the impact of critic selection and varying the number of candidates $N$ on performance across different datasets and environments. The reported values are Mean ± Standard Error over 5 training seeds × 500 episode seeds.

| Tasks | | Number of Candidates | | | | | | |
|---|---|---|---|---|---|---|---|---|
| Dataset | Environment | $N = 1$ | $N = 2$ | $N = 5$ | $N = 10$ | $N = 20$ | $N = 50$ | $N = 100$ |
| Medium-Expert | HalfCheetah | 96.9 ± 0.1 | 97.5 ± 0.1 | 98.0 ± 0.0 | 98.3 ± 0.0 | 98.5 ± 0.0 | 98.6 ± 0.1 | 98.7 ± 0.1 |
| Medium-Replay | HalfCheetah | 47.0 ± 0.0 | 47.9 ± 0.0 | 48.5 ± 0.0 | 48.7 ± 0.0 | 48.9 ± 0.0 | 48.9 ± 0.0 | 49.0 ± 0.0 |
| Medium | HalfCheetah | 51.2 ± 0.0 | 52.4 ± 0.0 | 53.5 ± 0.0 | 54.2 ± 0.0 | 54.8 ± 0.0 | 55.4 ± 0.0 | 55.8 ± 0.0 |
| Medium-Expert | Hopper | 85.9 ± 2.0 | 92.7 ± 1.9 | 92.4 ± 2.0 | 88.6 ± 2.0 | 83.1 ± 2.0 | 77.7 ± 1.9 | 74.0 ± 1.7 |
| Medium-Replay | Hopper | 101.8 ± 0.0 | 101.9 ± 0.0 | 102.0 ± 0.0 | 102.0 ± 0.0 | 102.0 ± 0.0 | 102.0 ± 0.0 | 102.0 ± 0.0 |
| Medium | Hopper | 98.5 ± 1.1 | 100.2 ± 0.8 | 102.5 ± 0.1 | 102.7 ± 0.1 | 102.9 ± 0.0 | 102.6 ± 0.2 | 102.7 ± 0.2 |
| Medium-Expert | Walker2d | 112.5 ± 0.0 | 112.7 ± 0.0 | 113.0 ± 0.0 | 113.1 ± 0.0 | 113.2 ± 0.0 | 113.4 ± 0.0 | 113.5 ± 0.0 |
| Medium-Replay | Walker2d | 101.9 ± 0.2 | 102.0 ± 0.0 | 102.0 ± 0.0 | 101.9 ± 0.0 | 101.7 ± 0.0 | 101.5 ± 0.0 | 101.5 ± 0.0 |
| Medium | Walker2d | 91.4 ± 0.3 | 91.8 ± 0.2 | 91.3 ± 0.1 | 91.1 ± 0.1 | 91.0 ± 0.1 | 90.6 ± 0.3 | 90.3 ± 0.4 |
| **Performance** | | 87.5 | 88.8 | **89.2** | 89.0 | 88.5 | 87.9 | 87.5 |
| Mixed | Kitchen | 66.6 ± 0.4 | 68.4 ± 0.4 | 69.8 ± 0.4 | 70.4 ± 0.4 | 70.6 ± 0.3 | 70.2 ± 0.4 | 70.3 ± 0.4 |
| Partial | Kitchen | 94.2 ± 0.6 | 95.5 ± 0.6 | 94.8 ± 0.6 | 94.2 ± 0.6 | 93.8 ± 0.6 | 93.2 ± 0.6 | 92.8 ± 0.6 |
| **Performance** | | 80.4 | 82.0 | **82.3** | 82.3 | 82.2 | 81.7 | 81.6 |
| Diverse | Antmaze-Large | 3.8 ± 0.8 | 42.3 ± 2.1 | 65.2 ± 2.0 | 70.0 ± 2.0 | 72.9 ± 1.9 | 71.7 ± 2.0 | 71.3 ± 2.0 |
| Play | Antmaze-Large | 77.8 ± 1.8 | 82.4 ± 1.7 | 81.7 ± 1.7 | 81.0 ± 1.7 | 80.4 ± 1.7 | 77.8 ± 1.8 | 73.4 ± 1.9 |
| Diverse | Antmaze-Medium | 92.1 ± 1.1 | 92.4 ± 1.1 | 88.8 ± 1.4 | 83.8 ± 1.6 | 78.8 ± 1.8 | 71.0 ± 2.0 | 64.7 ± 2.1 |
| Play | Antmaze-Medium | 88.0 ± 1.4 | 87.3 ± 1.5 | 85.3 ± 1.5 | 84.0 ± 1.6 | 84.2 ± 1.6 | 84.6 ± 1.6 | 82.3 ± 1.7 |
| **Performance** | | 65.4 | 76.1 | **80.3** | 79.7 | 79.1 | 76.3 | 72.9 |
| Large | Maze2D | 201.8 ± 1.9 | 203.5 ± 1.8 | 199.2 ± 2.0 | 194.3 ± 2.1 | 193.2 ± 2.1 | 187.7 ± 2.3 | 183.9 ± 2.4 |
| Medium | Maze2D | 143.9 ± 1.6 | 149.1 ± 1.5 | 150.1 ± 1.5 | 149.8 ± 1.5 | 148.4 ± 1.6 | 149.0 ± 1.6 | 146.4 ± 1.7 |
| Umaze | Maze2D | 138.3 ± 1.8 | 141.5 ± 1.7 | 144.3 ± 1.7 | 146.2 ± 1.7 | 143.5 ± 1.7 | 144.5 ± 1.6 | 145.0 ± 1.7 |
| **Performance** | | 161.3 | **164.7** | 164.5 | 163.4 | 161.7 | 160.4 | 158.4 |

## D.4. Visualizations of Habitual and Goal-Directed Policy Distributions

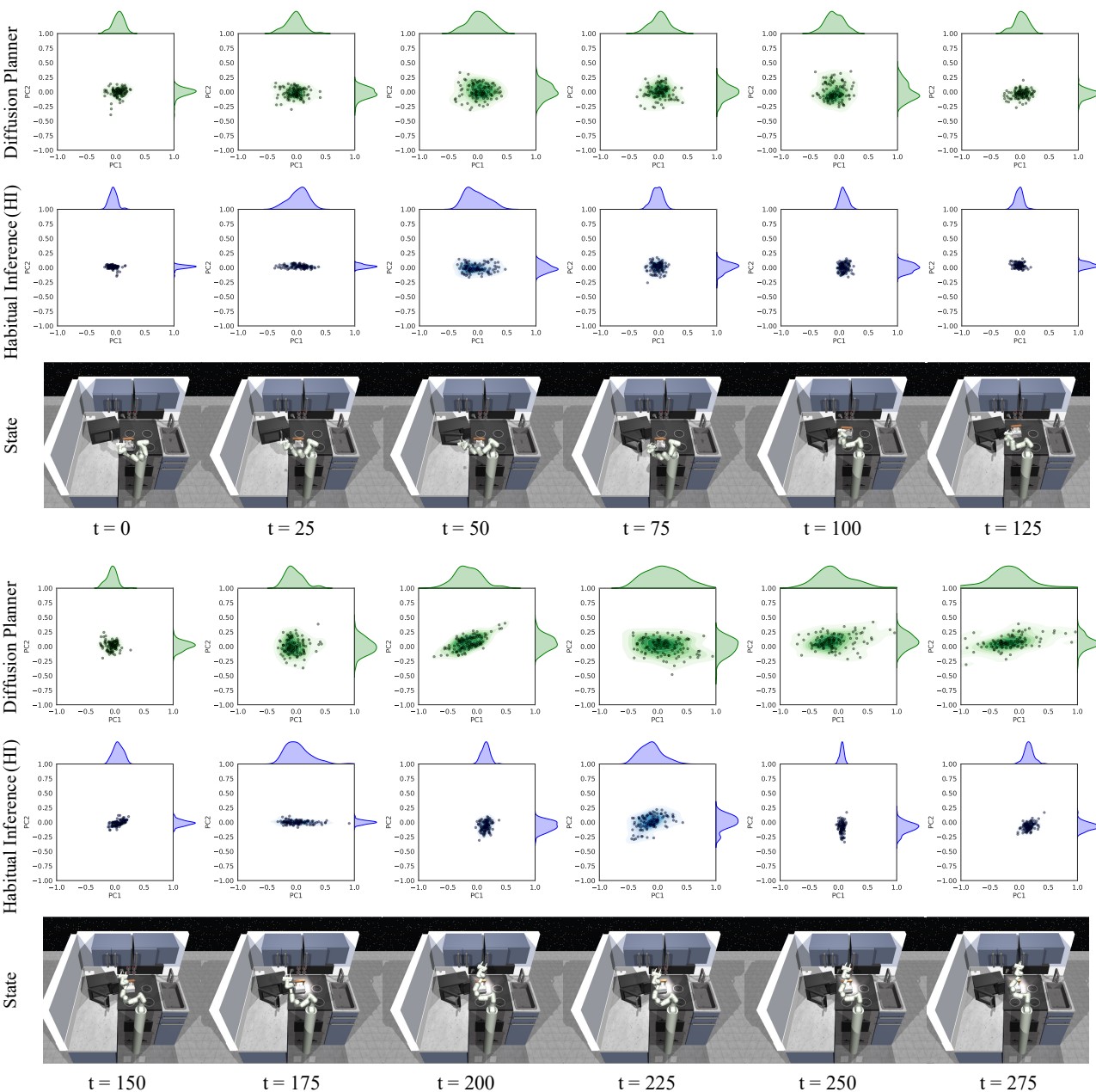

*Figure 9.* Action distribution visualization on Kitchen, plotted the same way as Figure 5. The actions are dimension-reduced by PCA.

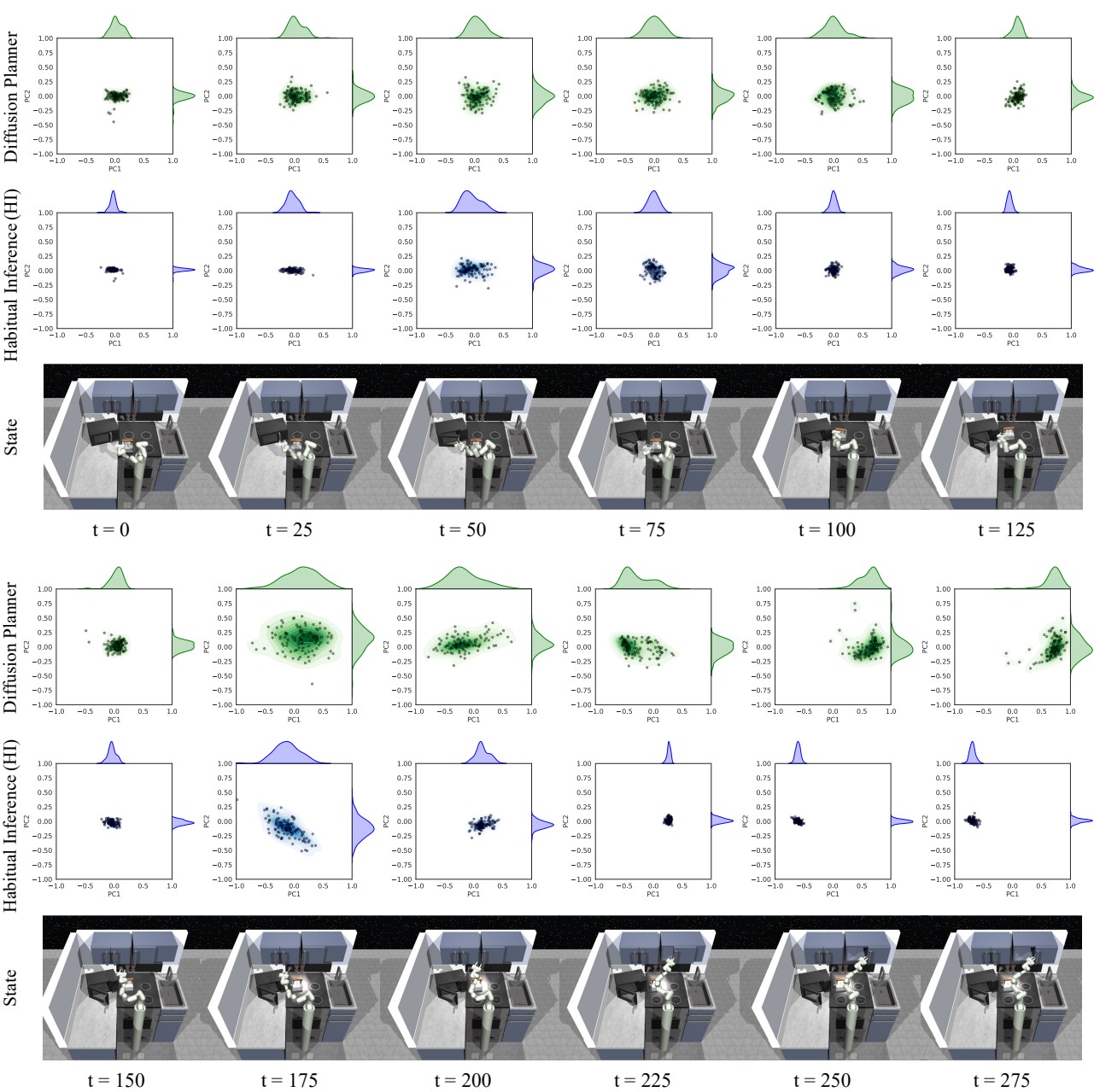

*Figure 10.* Another example of action distribution visualization on Kitchen, plotted the same way as Figure 5. The actions are dimension-reduced by PCA.

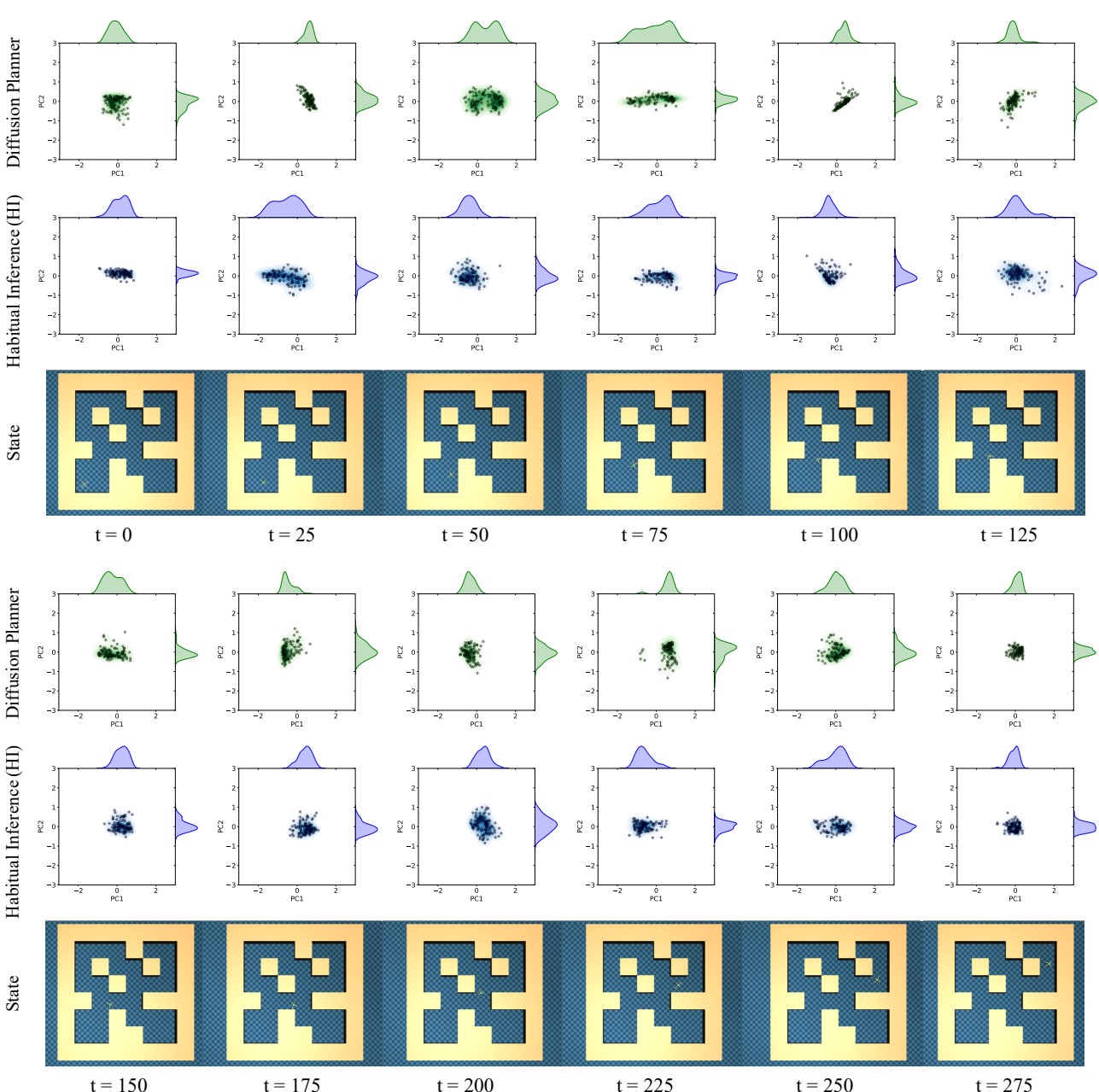

*Figure 11.* Action distribution visualization on AntMaze, plotted the same way as Figure 5. The actions are dimension-reduced by PCA.

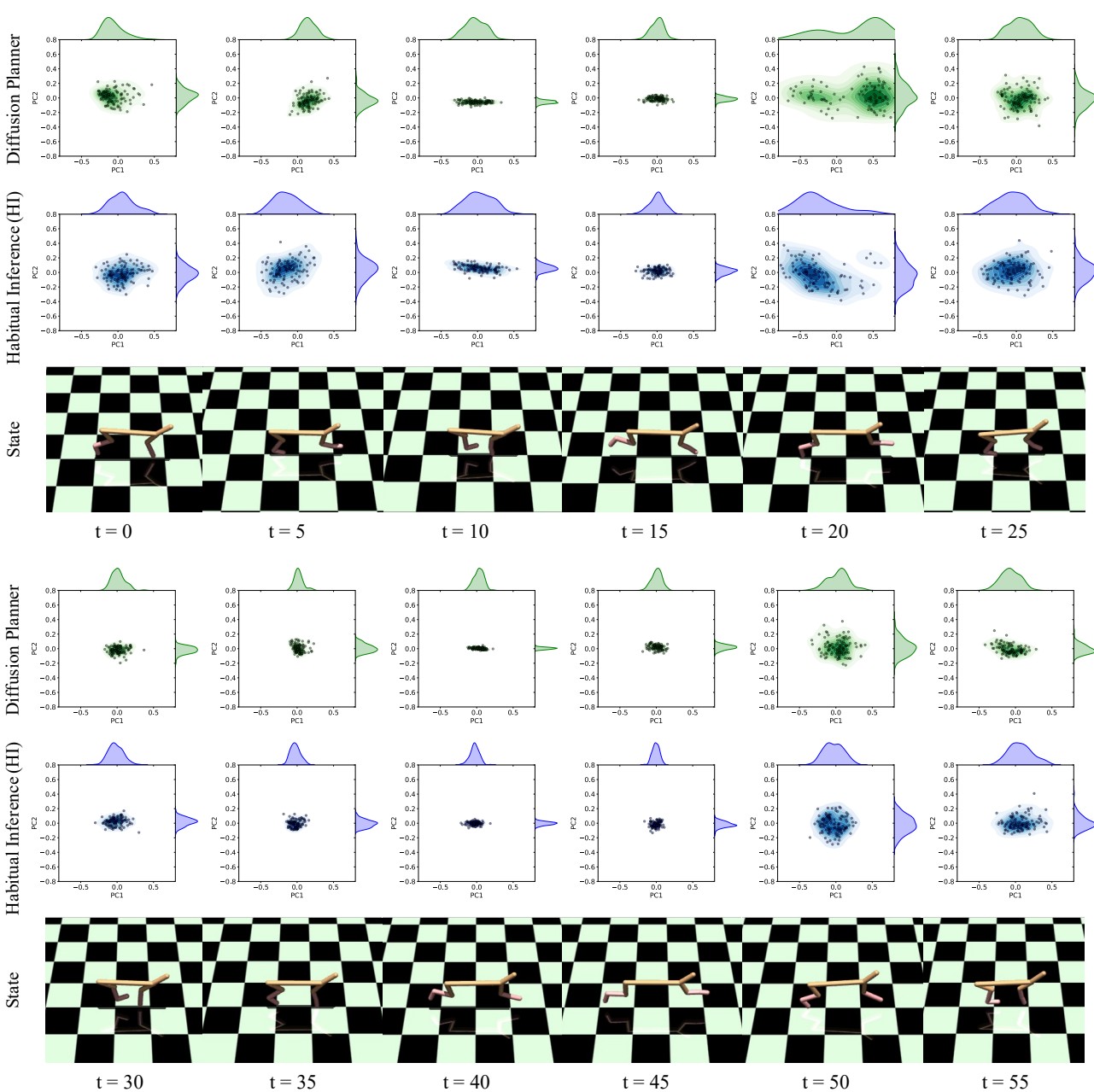

*Figure 12.* Action distribution visualization on MuJoCo, plotted the same way as Figure 5. The actions are dimension-reduced by PCA.

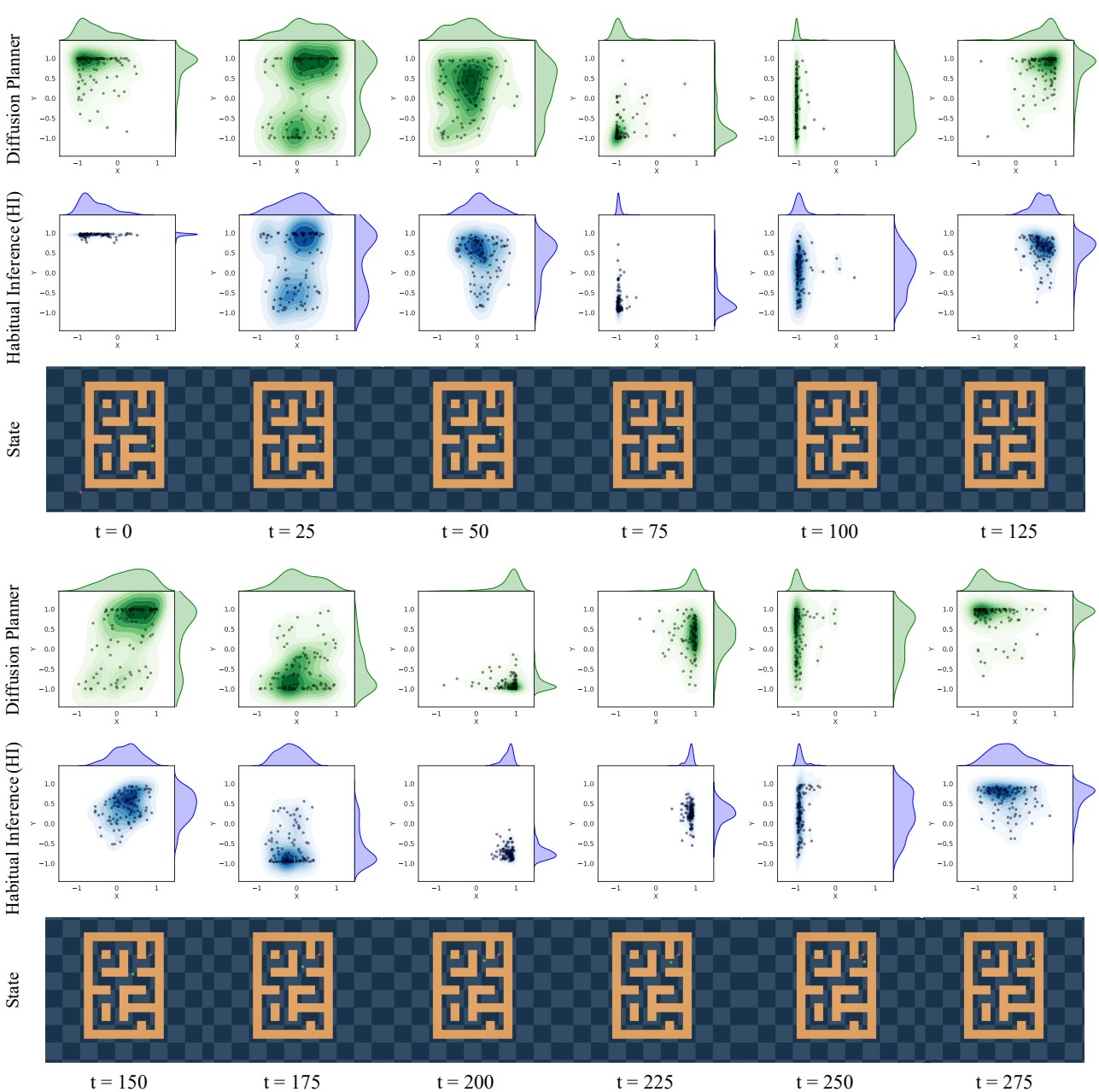

*Figure 13.* Another example of action distribution visualization on Maze2D, plotted the same way as Figure 5.

