# OpenReview forum: "Habitizing Diffusion Planning for Efficient and Effective Decision Making"
_ICML.cc/2025/Conference — ICML 2025 poster_

### Official Review · Reviewer_6Hyb · 2025-03-03

**Overall Recommendation:** 3

**Summary:**

This paper introduces a novel and general framework that can accelerate the existing diffusion-based planning models. The motivation is that, most of the diffusion-based planning methods are very slow due to the iterative denoising steps during deployment. In this work, a VAE-like learning framework is proposed to learn the distilled policy from the pre-trained diffusion models. The framework contains a prior encoder (state as input), a posterior encoder (state, action as input) and a latent decoder based on the posterior latent. The typical MSE reconstruction loss is used to distill the diffusion policy to the VAE-policy, and an extra KL-divergence loss is devised to align the decision space between prior and posterior encoders. A critic is trained to evaluate the quality of the samples. And during inference, out of the generated samples, the one with the highest critic score will be selected for deployment. The authors conduct extensive experiments under D4RL benchmark and compare with deterministic policies, diffusion policies/planners and accelerated diffusion-based methods. The proposed method demonstrates a much better computation efficiency with similar solution quality as the strongest baselines.

## update after rebuttal
I have read the author's response as well as other reviewers' comments. Overall, I think the work is novel and is okay to be accepted by ICML, though a few flaws exist. I updated my final ratings as "weak accept."

**Claims And Evidence:**

Yes.

**Essential References Not Discussed:**

No.

**Experimental Designs Or Analyses:**

Yes. I checked all the experiments.

**Methods And Evaluation Criteria:**

Yes.

**Other Comments Or Suggestions:**

N/A.

**Other Strengths And Weaknesses:**

Pros:
* The idea is very straightforward and novel.
* The paper is easy to read.
* Extensive experiments have been conducted (including reasonable ablation studies).

Cons:
* Missing vital baselines: should compare with flow-based methods [1, 2], vanilla VAE, and consistency models [3, 4].
* The proposed method requires a pre-trained diffusion model and also needs to train a critic - which requires more time in training than other "train-from-scratch" methods.
* Though the critic can improve per-step quality, the overall episode return does not seem to be improved much and even degrades later.

References:
[1] Zhang, Zhilong, et al. "Flow to better: Offline preference-based reinforcement learning via preferred trajectory generation." The Twelfth International Conference on Learning Representations. 2023.
[2] Zhang, Qinglun, et al. "FlowPolicy: Enabling Fast and Robust 3D Flow-based Policy via Consistency Flow Matching for Robot Manipulation." arXiv preprint arXiv:2412.04987 (2024).
[3] Chen, Yuhui, Haoran Li, and Dongbin Zhao. "Boosting continuous control with consistency policy." arXiv preprint arXiv:2310.06343 (2023).
[4] Prasad, Aaditya, et al. "Consistency policy: Accelerated visuomotor policies via consistency distillation." arXiv preprint arXiv:2405.07503 (2024).

**Questions For Authors:**

Why not report the computation time for BC and SRPO baselines?

**Relation To Broader Scientific Literature:**

This paper proposed diffusion model framework can work at very high frequency, opening the door for fast online deployment for systems that require real-time performance.

**Theoretical Claims:**

Yes (although there is no proof in this paper).

---

> ### Author Rebuttal · Authors · 2025-04-01
>
> Thank for reviewing our paper and acknowledging "***idea is very straightforward and novel / easy to read / extensive experiments / reasonable ablation studies.***" We believe the following response can address your concerns.
>
>
> > Q1: Comparison with other generative decision-making baselines
>
> Thank you for your suggestions. Since references 2 and 4 involve visuomotor policies and lack the same benchmarks, we compared our method with:
> - Flow to Better: Offline preference-based reinforcement learning via preferred trajectory generation
> - Boosting continuous control with consistency policy
>
> The policy of FTB is also deterministic, which is the same as BC/SRPO.
>
> | Environment | HI (Ours) | FTB | CPIQL | CPQL |
> |-------------|-----------|-----|-------|------|
> | HalfCheetah-ME | **98.0 ± 0.0** | 85.2 ± 0.7 | 81.0 ± 1.7 | 97.8 ± 0.5 |
> | HalfCheetah-MR | **48.5 ± 0.0** | 38.4 ± 1.3 | 48.0 ± 1.4 | 46.6 ± 0.8 |
> | HalfCheetah-M | 53.5 ± 0.0 | - | 54.6 ± 1.0 | **56.9 ± 0.9** |
> | Hopper-ME | 92.4 ± 2.0 | **111.1 ± 2.0** | 110.6 ± 1.4 | 110.4 ± 3.2 |
> | Hopper-MR | **102.0 ± 0.0** | 89.6 ± 4.9 | 100.6 ± 1.5 | 97.7 ± 4.6 |
> | Hopper-M | **102.5 ± 0.1** | - | 99.7 ± 2.0 | 99.9 ± 4.5 |
> | Walker-ME | **113.0 ± 0.0** | 109.3 ± 0.3 | 110.9 ± 0.2 | 110.9 ± 0.1 |
> | Walker-MR | **102.0 ± 0.0** | 79.1 ± 1.4 | 91.8 ± 2.8 | 93.6 ± 5.6 |
> | Walker-M | **91.3 ± 0.1** | - | 86.2 ± 0.6 | 82.1 ± 2.4 |
> | **Tasks Average** | **89.2** | - | 87.0 | 88.4 |
> | **Shared Tasks (w/o M dataset) Average** | **_92.7_** | 85.5 | 90.5 | **92.8** |
> | **Frequency** | 1329.7 | **11892.6** |  578.2 | 570.1 |
>
> We conducted comparisons on common MuJoCo locomotion tasks (since these work did not present the results on the other benchmarks in our paper, and FTB lacks the results on Medium dataset). Habi achieves the best performance in most tasks and demonstrates a higher overall score across environments.
>
> Importantly, we do not want to convey the message that Habi (variational Bayesian method) is better than flow-based method. Indeed, they are **orthogonal** and can be combined (e.g., Kingma et al., 2016. "Improved Variational Inference with Inverse Autoregressive Flow"). We appreciate your question and will have a deeper dive into potential collaboration between flow-based method in our future work.
>
>
> > Q2: The proposed method requires a pre-trained diffusion model and also needs to train a critic - which requires more time in training than other "train-from-scratch" methods.
>
> Thank you for your question. This work positions Habi as a general acceleration framework designed to speed up given diffusion planners. Our starting point is to maintain diffusion planning performance while significantly improving decision frequency.
>
> Indeed, Habi's training process is light-weighted,e.g., Habi only takes around 2 hours for Habitual Training using one A100 GPU.
>
> > Q3: Though the critic can improve per-step quality, the overall episode return does not seem to be improved much and even degrades later.
>
> There is probably a misunderstanding. We assume you are talking about Figure 7 (please let us know if not the case). Figure 7 illustrates the impact of increasing candidates on performance given the same critic. When there are too many candidates, performance may decline. Therefore, around 5 candidates is recommended. This phenomenon occurs because too many candidates amplify the critic's judgment errors for individual actions.
>
>
> > Q4: Why not report the computation time for BC and SRPO baselines?
>
> We acknowledge that BC and SRPO are super fast, as they used deterministic policy function with single-pass inference (i.e., an MLP). We did not report the computational time since the bottleneck of inference time no longer come from action computation, but rather from other parts such as `env.step(action)`. Thanks for the advice, and we will revise the paper to clarify this.
>
> However, probabilistic generative models like diffusion policies and planners typically achieve superior performance by modeling complex, multi-modal action distributions -- but at significant computational cost. Our work focuses specifically on accelerating probabilistic generative decision-making methods while maintaining their performance advantages over deterministic approaches. Habi addresses this trade-off by accelerating probabilistic generative models while preserving their stochastic nature and performance benefits.
>
>
> Thank you again for your constructive feedback! We hope the responses above have addressed all your comments. Please kindly let us know if you have additional suggestions, and we would be more than happy to discuss them.

---

### Official Review · Reviewer_ybn9 · 2025-03-13

**Overall Recommendation:** 3

**Summary:**

This paper presents Habi, a framework that habitizes diffusion planning into faster decision-making models by using a VAE-based approach inspired by biological habit formation. While the method demonstrates impressive speedups and maintains comparable performance to diffusion planners, the technical innovation is limited as it primarily applies standard VAE techniques to policy distillation without substantial modifications. Additionally, the claimed advantages over direct distillation are marginal in several tasks and lack thorough component-wise analysis. I have some concerns about the experimental setup and fairness of comparisons, as Habi fundamentally depends on pre-trained diffusion planners and should be more accurately positioned as an "acceleration technique" rather than a standalone decision-making algorithm.

**Claims And Evidence:**

1. I have some concerns about the experimental setup and fairness of comparisons. The paper frames Habi as a complete decision-making method, but its fundamental nature is a two-stage process that depends on pre-trained diffusion planners. This dependency is not adequately acknowledged, creating a misleading comparison. Habi should be more accurately positioned as an "acceleration technique" rather than a standalone decision-making algorithm. This would better reflect its true nature and enable more appropriate comparisons with other acceleration approaches.
2. For a truly fair evaluation, the paper should provide comprehensive cost analyses that include both the initial planner training and subsequent habitization process, giving readers a complete understanding of the total computational investment required. Moreover, the authors claim superiority over standard distillation approaches, but evidence is unconvincing. In several tasks (MuJoCo and Maze2D), performance improvements are marginal (only 2-5% better). Without comprehensive component-wise analysis, it's unclear what drives these improvements or whether they are statistically significant.

**Essential References Not Discussed:**

See above.

**Experimental Designs Or Analyses:**

1. The comparision and the experiment setup is not convicing.(See Claims Q1 and Q2)
2. While the paper includes analysis on the number of candidate samples (N), it lacks comprehensive ablations for other critical components such as network architecture choices, latent space dimensionality, and the impact of different diffusion planners as teachers.

**Methods And Evaluation Criteria:**

1. The comparision and the experiment setup is not convicing. The comparison between Habi (which utilizes pre-trained planners) and methods that learn from scratch (like AdaptDiffuser) is fundamentally imbalanced.(See Claims Q1 and Q2)
2. The "Direct Distill" baseline lacks detailed implementation specifications, making it impossible to verify whether it represents state-of-the-art distillation techniques. Comparisons with established knowledge distillation methods (e.g., Hinton's approach with soft targets) are conspicuously absent.
3. The paper introduces a Critic component for action selection, but Table 5 shows that in many environments, performance without the Critic (N=1 case) is already strong. This raises questions about the necessity of this additional component and complicates the architecture without clearly justified benefits.

**Other Comments Or Suggestions:**

See above.

**Other Strengths And Weaknesses:**

1. Innovation largely limited to applying existing techniques (VAE, critic-based selection) in a new context
2. Positioning as a biologically-inspired method appears to be primarily a narrative framing rather than a substantive technical innovation
3. The lack of real-world deployment testing or vision-based tasks limits confidence in practical applicability

**Questions For Authors:**

See above.

**Relation To Broader Scientific Literature:**

See above.

**Theoretical Claims:**

1. While the habitization process draws inspiration from cognitive science, the theoretical mapping between brain processes and the VAE framework is superficial. The paper does not sufficiently establish why ELBO optimization should effectively model the transition from goal-directed to habitual behavior.
2. The core technical contribution is essentially applying a standard VAE for policy distillation. The ELBO loss (L = L_recon + β_KL·L_KL) comes directly from standard VAE theory with minimal modification. The paper reinterprets this as "habitization" without substantial theoretical innovation.

---

> ### Author Rebuttal · Authors · 2025-04-01
>
> We greatly appreciate your detailed and comprehensive comments, the following responses are to address your concerns.
>
> >Q1: Position Habi as an "acceleration technique" rather than a standalone algorithm.
>
> Thank you for the clarification. Actually, we indeed position Habi as a general acceleration framework for diffusion planners in this paper. Our goal was creating an elegant framework that significantly improves decision frequency while maintaining effectiveness on SOTA diffusion planners through brief habitual training (1-2 hours).
>
> Our focus is on decision-frequency, with performance comparisons to demonstrate Habi's effectiveness.
>
> >Q2: Comparisons with established distill methods (e.g., soft targets)
>
> Diffusion models' probability distributions aren't directly accessible, making traditional distillation methods using cross-entropy loss (e.g., soft targets) inapplicable. This motivated Habi's development.
>
> We included comprehensive comparisons with two categories of acceleration approaches: numerical acceleration methods (DiffuserLite) and distillation-like frameworks (DTQL). The Direct Distill baseline uses the same architecture as Habi but performs imitation learning directly on state-action pairs.
>
> >Q3: Performance improvements seem marginal for simple tasks (Maze2D, MuJoCo)
>
> There is probably a misunderstanding. Our rigorous evaluation methodology (5 training seeds × 500 evaluation seeds) demonstrates these improvements are **_statistically significant_** and reproducible, which is supported by:
>
> $$\mu_{avg} = \frac{1}{G} \sum_{i=1}^G \mu_i$$
>
> $$stderr_{avg} =\frac{\sqrt{\frac{1}{G}\sum_{i=1}^G(\mu_i^2 + N_{seed} (stderr_i)^2) - \mu_{total}^2}}{\sqrt{G*N{seed}}},$$
> where $G$ is the task number, $N_{seed}$ is the number of seed.
>
> **Performance Comparison**:
>
> |Task|Method|HI(Ours)|HI w/o Critic|Direct Distill|
> |--|--|--|--|--|
> |MuJoCO|Absolute|89.24±0.29|87.46±0.32|84.98 ± 0.26
> |-|Δ vs Ours|-|-1.78 (p=1.88e-5)|-4.26 (p<1e-10)|
> |Maze2D|Absolute|164.53±1.07|161.33±1.11|159.87±1.10
> |-|Δ vs Ours|-|-3.20 (p=0.019)|-4.66 (p=0.001)|
>
> Moreover, the improvements becomes more pronounced in ***complex environments***: in Kitchen and Antmaze domains, Habi exhibits ***19.1–29.4%*** stronger performance.
>
> >Q4: Necessity of Critic.
>
> Thank you for your question. The critic is essential. Without it, performance in complex environments like Antmaze suffers an 18.5% degradation. Tested with over 500 seeds, the critic consistently provides performance improvements. Meanwhile, the critic is a shallow MLP that doesn't add significant computational burden.
>
> > Q5: Questions on network structures, diffusion planners, and latent dimensionality.
>
> Thank you for your suggestion. For high decision speed, MLPs are already sufficiently simple. DV and DQL adequately represent the two types of diffusion planning. Regarding latent dimensionality, our experiments show:
>
> |Env|dim(z)=64|128|256|512|
> |--|--|--|--|--|
> |Antmaze-L-D|62.2±2.1|60.7±2.0|65.2±2.0|70.6±2.0|
> |Antmaze-L-P|79.3±1.8|80.3±1.7|81.7±1.7|85.9±1.5|
> |Antmaze-M-D| 88.5±1.4|89.6±1.3|88.8±1.4|78.6±1.8|
> |Antmaze-M-P| 78.0±1.8|82.3±1.7|85.3±1.5|86.7±1.5|
> |**Avg**|77.0|78.2|80.3|80.5|
> |Maze2D-L|204.0±1.9|204.3±1.9|199.2±2.0|202.4±1.9|
> |Maze2D-M|151.1±1.5|151.8±1.4|150.1±1.5|149.2±1.6|
> |Maze2D-U|143.6±1.7|144.5±1.7|144.3±1.7|144.1±1.7|
> |**Avg**|166.2|166.9|164.5|165.2|
>
> Latent dimension has **_modest impact_** on performance. For Antmaze, there's slight improvement as dimension increases. For Maze2D, performance remains stable. As our method does not involve an information bottleneck, we selected dim(z)=256 as a balanced choice providing good performance without unnecessary computational overhead.
>
> >W1: Innovation & Q5: Difference between Habi and VAE
>
> We understand your concerns. As acknowledged by other reviewers (***qHBM***, ***JDXx***), Habi maintains simple and elegance for easy using, while Habi and VAE are different fundamentally:
>
> **Function**: VAE's latent dimension is smaller than input dimension, forming an information bottleneck; Habi uses a large latent dimension (256) as it doesn't target compact representation.
>
> **Latent Bottleneck**: VAE's latent dimension is smaller than input dimension, forming an information bottleneck; Habi uses a large latent dimension (256) as it doesn't target compact representation.
>
> **Learnable Prior**: VAE employs a fixed unit-Gaussian prior; Habi's prior distribution conditions on the current state and is learned.
>
> >W2. biologically-inspired rather than technical innovation
>
> Thanks for the suggestion. While the high-level idea of our paper is brain-inspired, we agree that we should more clearly highlight the technical innovations by revising the paper.
>
> >W3. Lack of real-world, vision-based deployment.
>
> Thank you for the suggestion. We acknowledge it in Section 6 of our paper. This work focuses on algorithmic contributions using standard offline RL for robust benchmarking in the simulation, which is common practice in the field.

---

### Official Review · Reviewer_JDXx · 2025-03-14

**Overall Recommendation:** 3

**Summary:**

This paper has introduced a simple yet effective framework to speed up Diffusion-based planners (Habi). During training Habi learns:
- A prior encoder for context (state)
- A Posterior encoder and decoder for distilling learned planning in diffusion.
- A critic for evaluating actions.
I like the elegant idea and strong performance, while there are a few questions/claims that I hope to discuss during the rebuttal period.

**Claims And Evidence:**

- Does the expert/teacher planner have to be a diffusion-based model?

From the framework, I didn't see any special assumptions on the pretrained planner, and it seems to me this should work for any learned model-based RL algorithm. What's the relationship between the proposed framework and diffusion-planners?

- The claim in L107-108 is a bit too strong:

"Habi can be used straightforwardly for any diffusion planning and diffusion policy models." While it seems that in the experiments the authors have only implemented HI for one base planner (correct me if I misunderstood).

**Essential References Not Discussed:**

N/A

**Experimental Designs Or Analyses:**

This is one design choice that I hope to see some analyses:
- Why is the framework using $z_t^q$ instead of $z_t^p$ for training the Critic?

It seems to me that during inference, $z_t^p$ is the one used for Critic evaluation, so using $z_t^p$ during training seems to be a more intuitive design, is there any theoratical/empirical support for such design?

**Methods And Evaluation Criteria:**

The method and evaluation make sense for the problem and application.

**Other Comments Or Suggestions:**

N/A

**Other Strengths And Weaknesses:**

Strengths:
- Simple yet effective approach
- Well-motivated idea
- Well written paper

Weaknesses:
- Some claims are too strong to be supported in the experiments.
- It is unclear to me about the relationship between this approach and diffusion.

**Questions For Authors:**

N/A

**Relation To Broader Scientific Literature:**

N/A

**Theoretical Claims:**

N/A

There is only a (widely known) ELBO theoretical proof for VAE shown in the supplementary.

---

> ### Author Rebuttal · Authors · 2025-04-01
>
> We appreciate your thoughtful comments and acknowledging "***Simple yet effective approach / Well-motivated idea / Well written paper / elegant idea and strong performance***". We will address all of your concerns as follows.
>
> > Q1: Does the expert/teacher planner have to be a diffusion-based model? What's the relationship between the proposed framework and diffusion-planners?
>
> Good point! Theoretically, Habi also supports acceleration for other ***probabilistic generative*** decision-making models. However, Habi is especially well-suited for diffusion planners for several important reasons:
>
> 1. Many other generative models (like VAEs or flow-based models) are already computationally efficient, but often face performance limitations rather than speed constraints [1, 2]. Their bottleneck, however, is typically quality of decisions, not decision frequency.
> 2. Diffusion models have emerged as the most powerful and well-performing probabilistic generative models in decision-making, as evidenced by the impact of Diffuser and related work. However, diffusion planners are inherently slow during inference due to their multi-step denoising process, which limits their practical application in real-world scenarios.
>
> Habi addresses this crucial gap between 1) and 2) by accelerating diffusion-based decision-making models while maintaining nearly lossless performance. To our best knowledge, Habi is the first framework to successfully preserve diffusion models' superior performance while achieving decision speeds comparable to traditional generative models.
>
> We appreciate your suggestion and will clarify this important distinction in the revised paper.
>
> > Q2: The claim in L107-108 is a bit too strong:
> "Habi can be used straightforwardly for any diffusion planning and diffusion policy models." While it seems that in the experiments the authors have only implemented HI for one base planner (correct me if I misunderstood).
>
> Actually, we used Habi on two base diffusion decision making models: [1] and [3] (line 738, Appendix Table 3). We selected these two models since they are best performing on corresponding benchmarks.
>
> However, we greatly appreciate this suggestion since we did not realize this clarity issue. We will make more clear statements in the main texts of the revised paper.
>
> > Q3: Why is the framework using ztq instead of ztp for training the Critic?
>
> The Prior distribution and Posterior distribution are constrained through KL divergence. Due to the nature of the KL constraint KL(q||p):
>
> $$KL(q(z) \parallel p(z)) = \int q(z) \log \frac{q(z)}{p(z)} dz$$
>
> The prior $p(z)$ is required to cover the posterior distribution $q(z)$, but points with low probability in the posterior may still be sampled by the prior. Using these low-probability samples to train the critic would introduce noise and potentially mislead the training process.
>
> Therefore, using latents generated from the posterior can reduce the impact of these misleading samples on critic training, ensuring more reliable and accurate critic learning. Intuitively, this shares a roughly similar idea with *Teacher Forcing* in autoregressive models: During training, instead of feeding the model's own predicted output (akin to prior z in Habi), teacher forcing uses the ground-truth (akin to posterior z in Habit) to prevents compounding errors.
>
> We also make an additional experiment on maze2d to empirically validate this design choice:
>
> |Environment| Posterior Critic | Prior Critic |
> |----|----|----|
> |Maze2D-Large| 199.2 ± 2.0 | 195.3 ± 2.2 |
> |Maze2D-Medium| 150.1 ± 1.5 | 150.6 ± 1.5 |
> |Maze2D-Umaze| 144.3 ± 1.7 | 143.0 ± 1.7 |
> |**Average**| **164.5** | 163.0 |
>
> It can be seen that using posterior z for critic training works slightly better.
>
> Thank you again for your acknowledgement of Habi and your contribution to our paper. Hope that we have address your concerns.
>
> **Reference**
>
> [1] What Makes a Good Diffusion Planner for Decision Making? Lu et al. ICLR 2025
>
> [2] CleanDiffuser: An Easy-to-use Modularized Library for Diffusion Models in Decision Making, Dong et al. NeurIPS2024
>
> [3] Diffusion policies as an expressive policy class for offline reinforcement learning, Wang et al. ICLR 2023
>
> Thank you again for your constructive feedback! We hope the responses above have addressed all your comments. Please kindly let us know if you have additional suggestions, and we would be more than happy to discuss them.

---

### Official Review · Reviewer_qHBM · 2025-03-15

**Overall Recommendation:** 3

**Summary:**

This paper proposes the Habi algorithm (a Diffusion Planner), which combines excellent performance with high-frequency inference speed. It utilizes a VAE-like inference framework to distill information from the diffusion planning process. Extensive experiments were conducted across various environments, achieving consistent improvements.

**Claims And Evidence:**

yes

**Essential References Not Discussed:**

I think some mbrl work should be discussed, see weakness.

**Experimental Designs Or Analyses:**

yes

**Methods And Evaluation Criteria:**

yes

**Other Comments Or Suggestions:**

see above

**Other Strengths And Weaknesses:**

pro:
- The paper is written very clearly, and the research question is very important. I quite agree that decision speed is a critical issue in the application of diffusion models for decision making.
- The method makes sense and is supported by extensive experimental validation. It compares a series of strong baselines, such as diffusion planners and diffusion policies.
- The method is simple and elegant, and it is expected to achieve both efficiency and high performance.

cons:
- Why not use a GPU for decision-making speed tests? Why does the decision frequency of DiffuserLite seem to be significantly lower than the paper version?
- Although the article describes the core idea using habitual decision-making, is the overall concept closer to some world model and representation-related works, such as TDMPC, TDMPC2, and MRQ (Towards General-Purpose Model-Free Reinforcement Learning)? By achieving better representations, it enables the distillation of the diffusion planner, thereby achieving better performance. From this perspective, Habi is similar to a more advanced offline MBRL method. Could it be discussed in relation to advanced offline MBRL methods? For example, MOREC (Reward-Consistent Dynamics Models are Strongly Generalizable for Offline Reinforcement Learning)."
- Can the ideas of habi be applied to more realistic robot control diffusion policy or robot control environments (such as libero, rlbench, etc.)?
- Why is the diffusion planner algorithm rarely applied in real robotic control scenarios? As far as I know, most mainstream algorithms are based on diffusion policy.

**Questions For Authors:**

see above

**Relation To Broader Scientific Literature:**

n/a

**Theoretical Claims:**

yes

---

> ### Author Rebuttal · Authors · 2025-04-01
>
> We are grateful to your thoughtful comments and acknowledging our work "***written very clearly / research question is very important / method makes sense / supported by extensive experimental validation / simple and elegant***" as well as bringing the questions. We address your questions as follows.
>
> > Q1: GPU for decision-making speed tests.
>
> Thanks for the suggestion. In fact, we already provided the results on both CPU and GPU in Appendix Table 4. We reported the results on CPU as CPUs are more portable (as descent CPUs can be put into mobile phones), thus be more practical when considering local deployment of decision models on individual robots and edge devices.
>
> > Q2: Decision frequency of DiffuserLite
>
> We also noticed this. DiffuserLite serves as a key baseline for decision-making acceleration. We are unsure if absolute value differences stem from hardware variations, as we follow its [official code](https://github.com/diffuserlite/diffuserlite.github.io) for testing. In our experiments, we used consistent hardware (Apple M2 Max, Nvidia A100, AMD EPYC 7V13 64-Core) and software (PyTorch=2.2.2, numpy=1.22.4) to ensure fair and consistent speed test results. Therefore, the relative decision frequency presented in our paper could fairly reflects the differences in decision speed among various generative decision-making methods.
>
> > Q3: Relationship with world model and MBRL
>
> Thank you for the suggestion. We treat Habi purely as an acceleration framework for generative decision-making. Its training process does not involve Q-Learning, nor does it interact with virtual reward dynamics. Habi's goal is to maintain the performance of the original generative model while accelerating decision-making within 1-2 hours of Habitual Training. Indeed, Habi is not limited to accelerating model-based planners (e.g., Diffuser, DV) but can also accelerate model-free planners (e.g., IDQL, DQL). We appreciate your suggestion and will discuss Habi's connection to offline MBRL methods (e.g., MOREC) in the related works section.
>
> > Q4: Can the ideas of habi be applied to more realistic robot control or robot control environments?
>
> Yes, and thanks for suggesting extensive benchmarks. Beyond the Franka robotic arm manipulation tasks, we  evaluate Habi on the Adroit environment (including opening the door, driving the nail, repositioning the pen orientation, and relocating the ball) for dexterous manipulation assessment.
>
> | Robotics Environments | Diffusion Planner [1] | Habi |
> |--------------|------------------------|------|
> | Door | 104.2 ± 0.7 | 105.6 ± 0.3 |
> | Hammer | 124.4 ± 1.8 | 129.4 ± 0.4 |
> | Pen | 122.2 ± 1.8 | 121.0 ± 2.4 |
> | Relocate | 109.3 ± 0.5 | 108.8 ± 0.5 |
> | Franka Kitchen-M | 73.6 ± 0.1 | 69.8 ± 0.4 |
> | Franka Kitchen-P | 94.0 ± 0.3 | 94.8 ± 0.6 |
>
> The results demonstrate that Habi maintains comparable performance to diffusion planners across various robot control tasks. In some environments like Door, Hammer, and Kitchen-P, Habi even shows slight improvements. We believe this highlights the robustness of our simple-yet-effective method and its potential for real-world robotic applications.
>
> > Q5: Why is the diffusion planner algorithm rarely applied in real robotic control scenarios? As far as I know, most mainstream algorithms are based on diffusion policy.
>
> We first would like to make a disambiguity statement about the "diffusion policy" in your question.
>
>
> ***--If you meant the paper Diffusion policy: Visuomotor policy learning via action diffusion from Chi et al.***
>
>
> The premise of this question contains a misconception. Diffusion Policy is actually built upon Diffuser (a classic diffusion planner), with slight modifications to action modeling and visual inputs. Consequently, ***diffusion planner algorithms have indeed been deployed in robotics***, with DP being among the first to bridge the ML-robotics gap and demonstrate results on physical robots.
>
> Furthermore, the subsequent development directions of these two approaches differ: the robotics community typically emphasizes ***Imitation Learning*** with ***expert demonstrations***, while the ML community focuses on ***Offline RL*** that can derive optimal policies from ***varied-quality data***. The core algorithms, however, share the same diffusion-based foundation.
>
>
> ***--If you meant the model-free diffusion policies such as Diffusion Policies as an Expressive Policy Class for Offline Reinforcement Learning from Wang et al.***
>
> We believe one of the most crucial problems of diffusion planners are its heavy computation cost even on a descent GPU: As we have shown in Appendix Table 4, Diffuser take 0.3~0.6 second for one decsion, which is not acceptable to real-world robot. However, research shows diffusion planners can outperform diffusion policies on many tasks [1]. This is why our work focused on accelarating diffusion planning while maintaining its effectiveness.
>
> [1] What Makes a Good Diffusion Planner for Decision Making? Lu et al. ICLR 2025

---

### Decision · Program_Chairs · 2025-05-01

**Decision:**

Accept (poster)

**Comment:**

This paper introduces a method for accelerating diffusion-based planners. The reviewers appreciated the importance of the problem, the simplicity of the method, and the strength of the results. They also found the paper well written and well motivated. The main concerns (missing baselines, clarifying the computational costs) were addressed during the rebuttal period, alongside a number of other concerns. After the rebuttal and a discussion among the reviewers, all four reviewers agreed that the paper should be accepted. I am therefore recommending that the paper be accepted.

My expectation is that authors will incorporate the discussion and new experiments from the rebuttal into the camera ready version of the paper. I'd also encourage them to clarify the details for the "Direct Distill" baseline, as suggested by Reviewer 6Hyb.